# Highlighting Lactic Acid Bacteria in Beverages: Diversity, Fermentation, Challenges, and Future Perspectives

**DOI:** 10.3390/foods14122043

**Published:** 2025-06-10

**Authors:** Zahra S. Al-Kharousi

**Affiliations:** Department of Food Science & Nutrition, College of Agricultural and Marine Sciences, Sultan Qaboos University, Al-Khod 123, Muscat P.O. Box 34, Oman; umohaned@squ.edu.om; Tel.: +968-2414-1219

**Keywords:** beverages, bioactive compounds, bioremediation, drug delivery, fermentation, food safety, kefir, lactic acid bacteria, non-dairy, wine

## Abstract

Lactic acid bacteria (LAB) have long been recognized for their versatility and historical significance, with a remarkable capability to produce a wide range of bioactive compounds that can be used across food, pharmaceuticals, nutrition, agriculture, and sustainable industrial sectors. This review aims to explore the current state of knowledge regarding LAB in beverages, emphasizing their diversity across dairy, non-dairy, and hybrid beverage matrices. Key aspects discussed include fermentation processes, associated challenges, and future perspectives. By examining a wide array of studies, this review offers a holistic perspective on the role of LAB in influencing sensory characteristics (both desirable and undesirable), promoting health benefits, extending shelf life, and enhancing their safety. Furthermore, emerging trends are highlighted, such as the use of LAB for the development of novel LAB-based beverages, their use for bioremediation of toxic compounds, genetic engineering of LAB strains to optimize and tailor their fermentation outcomes, and their use in drug delivery.

## 1. Introduction

Lactic acid bacteria (LAB) are Gram-positive, non-spore-forming, generally non-motile bacteria classified under the order Lactobacillales (phylum Firmicutes). They exhibit cocci or rod shapes, thrive in both aerobic and anaerobic conditions, and play key roles in fermentation, with applications in food, agriculture, and health sectors [1,2,3,4]. Although LAB have been used for the production of fermented foods since ancient times, the acknowledgement of the health benefits related to their use in these foods was initiated at the beginning of the twentieth century [5]. Since then, the consumption of functional drinks produced by the action of LAB has been increasing [6,7], leading to the expansion of LAB-containing foods in the market [5] after their first industrial utilization in the 1930s [8]. This expansion has been driven by the growing consumer awareness and positive perception towards probiotics in beverages and other food products, as seen in Malaysia, where over 80% of consumers are knowledgeable about the benefits of probiotics [9]. Currently, LAB-containing foods are widely available in global markets, including China, Germany, Jordan, Korea, Lithuania, New Zealand, Poland, Singapore, Thailand, Turkey, and Vietnam. Probiotic strains have been used in these markets for decades without adverse events [10]. The global LAB market was valued at USD 1.16 billion in 2024 and is projected to reach USD 2.22 billion by 2034, growing at a CAGR (Compound Annual Growth Rate) of 6.1% [11]. The Asia-Pacific region is leading the global market and is expected to grow at a CAGR of 9.2%, driven by a strong cultural desire towards fermented food and beverages and increasing health consciousness in countries like China, Japan, and South Korea. The regions of North America and Europe are showing steady growth, with North America projected at a CAGR of 7.5% and Europe at 7.8%, supported by high consumer awareness and a strong focus on health and wellness products [12].

Moreover, the utilization of LAB allowed the innovation of a wide variety of products capable of meeting the consumers’ needs for healthy products such as fermented plant-based beverages and functional fruit juices [8,13,14,15]. Due to their powerful metabolic pathways and enzyme systems, LAB play a crucial role in the production, flavor enhancement [16,17,18], microbial stability [19], nutrition, and functionality [20] of many fermented dairy and non-dairy food products. The LAB enzymatic system permits them to be involved in glycolysis (sugar fermentation), lipolysis (fat degradation), and proteolysis (protein degradation) [3]. Therefore, LAB can ferment food macromolecules and decompose indigestible polysaccharides into valuable products, mainly lactic acid, thereby acidifying the food products. They also have the potential to transform undesirable substances and flavor compounds and produce various components during their metabolism, such as bacteriocins, vitamins, exopolysaccharides, short-chain fatty acids, and amines [3,21,22].

Our knowledge about the health benefits of LAB is expanding due to the probiotic properties of many strains. Examples of these benefits include reduction of lactose intolerance and cholesterol levels, vitamin synthesis, and enhancement of gut health [4,21,23]. LAB help maintain the balance of intestinal microflora by suppressing harmful bacteria, which is crucial for gut health. LAB fermentation enhances antioxidant activity in functional drinks. This is achieved through the production of bioactive compounds, such as phenolics and flavonoids [24]. They also have immunomodulatory and antitumor properties [25]. LAB produce bacteriocins and other antimicrobial compounds that inhibit the growth of pathogenic bacteria, contributing to the safety and shelf life of the functional drinks [14]. Moreover, LAB may help reduce cardiovascular diseases by producing bioactive compounds that have beneficial effects on heart health [26]. LAB-fermented drinks can play a role in managing metabolic diseases through their anti-inflammatory effects and ability to balance microbiota [27]. The level of LAB should be 10^6^ CFU/mL/g or more to exhibit the probiotic or nutraceutical functional properties [28,29].

Although about 80% of LAB use is in the dairy industry, a recent trend in their use is in plant-based products, where LAB can perform various functions. They can produce acidic metabolites during the fermentation process, which can protect active substances such as phenols and vitamins or convert them into other polyphenols, such as catechins and anthocyanins, resulting in fermented products with enhanced functionality, such as antioxidant activity [20]. LAB can transform complex, indigestible proteins, cellulose, and other substances in plant-based materials into amino acids, organic acids, volatile substances, and other active ingredients, thereby improving the digestibility and flavor of fermented products [30]. Moreover, LAB can be antagonistic to most spoilage and pathogenic microbes, as they can use the rich nutrients in plant-based materials to produce organic acids, bacteriocins, and other active ingredients that increase the stability and shelf life of food products [19,20]. LAB can also efficiently transform agricultural waste into value-added products [31], contributing to sustainability and green chemistry.

In dairy fermentation, the enzymatic activities of LAB include glycolysis, proteolysis, lipolysis, and the formation of various flavor compounds. Through glycolysis, LAB convert lactose and other sugars into lactic acid, reducing pH and acting as a preservative. For example, *Streptococcus thermophilus* and *Lactobacillus delbrueckii* ssp. *Bulgaricus* are commonly used in dairy fermentations and have specific growth rates on different sugars [32]. The proteolytic enzymes of LAB break down milk proteins into peptides and amino acids, contributing to flavor and texture, which is essential for the development of cheese and yogurt, where casein is hydrolyzed [33]. LAB lipolysis involves the breakdown of milk fats into free fatty acids and other compounds, influencing flavor [33,34]. Moreover, LAB produce diacetyl, acetoin, and other volatile compounds through citrate metabolism, enhancing the flavor of products like cultured butter and sour cream [34,35].

The enzymatic activities of LAB in plant-based products include carbohydrate fermentation, phytase activity, proteolytic activity, and enzyme inhibition. LAB ferment simple sugars like glucose and sucrose, but not polysaccharides, due to the lack of hydrolytic enzymes. Specific strains like *Leuconostoc mesenteroides* rapidly ferment sucrose, while others like *Lactococcus lactis* metabolize maltose and lactose [32,36]. Some LAB strains can improve mineral bioavailability in plant-based substrates due to their phytase activity that breaks down phytates, which are anti-nutrient substances [37]. LAB can also hydrolyze plant proteins, producing bioactive peptides with health benefits. For example, *Lacticaseibacillus rhamnosus* showed high proteolytic activity in soy milk and produced soy bioactive peptides, which are beneficial to health [38]. LAB can synthesize compounds like γ-aminobutyric acid (GABA) from plant substrates, contributing to the health benefits of fermented beverages [13]. In addition, fermentation can increase the content of phenolic substances and antioxidants in plant-based beverages. LAB fermentation can result in the production of compounds that inhibit enzymes like α-glucosidase and α-amylase, which are beneficial for managing blood sugar levels [39].

Given the rapid growth of the functional beverage sector, this review aims to summarize the current state of knowledge regarding LAB in beverages, emphasizing their diversity across dairy, non-dairy, and hybrid beverage matrices. This comparative approach helps readers understand how LAB behave and contribute across different substrates. Details about LAB fermentation processes, associated challenges, and prospects are also discussed to assist in bridging knowledge gaps and identifying research opportunities in the rapidly growing functional beverage sector. Furthermore, it discusses their possible contribution to the fermentation process, emphasizing the necessity to explore the LAB present in unexplored food matrices and utilize them to manufacture novel beverages and foods, possibly valorizing low-value raw materials.

## 2. Diversity and Function of Lactic Acid Bacteria in Beverages

As summarized in Table 1, LAB are involved in diverse life sectors, necessitating continuous identification of new potential strains and characterizing their classification and application in diverse fields of food, industry, and health [25]. This is because the metabolic byproducts of LAB that significantly affect and refine the sensory characteristics of the final food product are strain-specific [40]. Recently, the classification of LAB has been modified according to their genetic relationships, physiology, ecology, metabolic properties, amino acid identity, and other characteristics. The family Lactobacillaceae now includes 31 genera. The genus *Lactobacillus* was split into 25 genera, including 1 retained genus (*Lactobacillus*) and 1 previously existing genus (*Paralactobacillus*), with 23 new genera introduced to accommodate species formerly classified under *Lactobacillus*. The main genera of LAB include *Lactobacillus*, *Lactococcus*, *Streptococcus*, *Pediococcus*, *Weissella*, *Leuconostoc*, *Carnobacterium*, *Aerococcus*, *Enterococcus*, *Tetragenococcus*, *Oenococcus*, and *Vagococcus* [2,16,17,18,41].

### 2.1. Fermented Dairy Beverages

#### 2.1.1. Milk Kefir

Kefir is believed to have originated in the mountains of Tibet thousands of years ago. The word kefir means “living well” or “well-being”, reflecting the health benefits it offers to consumers. Kefir should contain at least 0.6% lactic acid, 2.7% protein, and less than 10% fat. The total microbial count should be at least 10^7^ CFU/mL, and the yeast count not less than 10^4^ CFU/mL. As opposed to water kefir, milk kefir is produced by fermenting bovine milk using milk kefir grains that are small, white or creamy in color, have a cauliflower and granular shape, and contain mostly kefiran exopolysaccharide. They contain LAB, AAB, yeasts, and filamentous fungi present in a symbiotic relationship. The kefir grains can be recovered after fermentation and reused for the next kefir fermentation [54]. They can also increase in biomass up to 3.5% after the fermentation process [55]. Many kefir microbiota have probiotic potential, such as resistance to high acidity and bile, the ability to adhere to the intestinal epithelial cells, the production of antagonistic molecules, and the reduction of pathogenic bacteria adhering to the gut cells. Beyond gut health, kefir can improve skin health, eczema, atopic dermatitis, burns, healing of scars, and rejuvenation [54]. *Lentilactobacillus kefiri*, *Lacticaseibacillus kefiranofaciens*, *Lacticaseibacillus kefirgranum*, *Lentilactobacillus parakefiri*, *S. thermophilus,* and *Lactococcus* bacteria are the predominant bacteria in milk kefir grains. This product is not suitable for vegans and those with lactose intolerance or allergy to milk proteins [55,56]. The sensory properties of milk kefir are attributed to the presence of acetaldehyde, acetoin, carbon dioxide, lactic acid, low levels of ethanol, and other aroma compounds in the final product [40].

LAB release several bioactive compounds in kefir beverages. Kefiran is a major exopolysaccharide produced by LAB such as *L. kefiranofaciens* in kefir, known for its antimicrobial properties. Exopolysaccharides were shown to inhibit pathogenic bacteria, such as *Listeria monocytogenes* and *Salmonella* Enteritidis [57]. LAB produce bacteriocins and organic acids, such as lactic acid and acetic acid, that contribute to the antimicrobial properties of kefir. LAB also produce bioactive compounds due to their proteolytic activity on milk proteins (caseins and whey proteins). These peptides exhibit various health benefits, including antimicrobial, antihypertensive, and immunomodulatory effects [58,59]. LAB in kefir also synthesize certain vitamins and release amino acids during fermentation, enhancing the nutritional value of the beverage [59]. Moreover, they produce volatile organic compounds, such as acetaldehyde, ethanol, diacetyl, and acetoin, contributing to the unique flavor and aroma of kefir [60].

Recently, innovative kefir beverages have been produced using non-conventional ingredients and new technologies. Incorporating fruit juices, plant extracts, and essential oils into kefir can enhance its antioxidant and functional properties. For instance, black carrot juice has been shown to produce water kefir-like beverages with high antioxidant activity and favorable sensory properties [61]. Likewise, a novel kefir beverage using date syrup, whey permeates, and whey has been developed, optimizing the formulation to achieve high antioxidant activities and acceptable organoleptic properties [62]. Moreover, some researchers formulated water kefir with Russian olive juice and optimized the process of production to maximize the phenolic content, antioxidant activity, and microbial viability to enhance the probiotic properties [63]. To meet consumers’ preferences, coffee-flavored kefir has been developed, and it showed promising results in terms of sensory acceptance and purchase intent, with high probiotic counts and antioxidant capacity [64]. Technological innovations have also been utilized in kefir production. For example, spray drying and encapsulation improved the stability and shelf life of kefir and maintained its functional properties. However, these techniques may impact the viability of beneficial microorganisms [65]. Moreover, kefir-containing snacks have been produced using 3D food printing, which could increase kefir consumption by offering attractive shapes and maintaining high probiotic content [66].

#### 2.1.2. Buttermilk

Butter churning results in the production of a liquid known as buttermilk with unique physicochemical characteristics, making it suitable for various applications in the food industry, such as sauces, confectionery, and bakery, as it improves the product’s water-holding capacity, texture, and lipid oxidation. Buttermilk has a lower concentration of triglycerides than whole milk but about seven times higher levels of phospholipids. It has naturally concentrated milk fat globule membranes. These bioactive phospholipids and proteins have been found to have antitumor and cholesterol-lowering potential [67]. Buttermilk is a good source of protein, calcium, phosphorus, riboflavin, and potassium, with a content of lactose and ash similar to whey [68]. A previous study found that the fermentation of buttermilk with LAB decreases the immunoreactivity of milk proteins, making it a healthier alternative for those with allergies to milk proteins. The composition of volatile fatty acids and phospholipids was also improved by LAB fermentation. For example, the content of L-α phosphatidylinositol has doubled from 5.2 to 10.48% of total phospholipids (PL), and the concentration of L-α phosphatidylcholine has significantly increased from 19.7 to 22.8% PL. Likewise, the concentration of acetic acid has increased from 90.6 to 99.9 mg/100 g dry matter (DM), and the content of butyric acid reached 74.1 mg/100 g DM, while it was not detected in control samples. The used bacteria included 31 LAB (genera: *Lactobacillus*, *Lactococcus*, and *Streptococcus*) and *Bifidobacterium* strains [67].

Thus, fermenting milk with different LAB species enhances its nutritional value, provides various health benefits, ensures safety through antimicrobial activity, and can extend the shelf life with proper management of post-acidification. Among the bioactive compounds produced is γ-aminobutyric acid, in which the LAB species such as *L. lactis*, *L. rhamnosus*, and *Lacticaseibacillus paracasei* can significantly increase content in fermented milk [69]. LAB species like *Lactobacillus delbrueckii* and *S. thermophilus* can produce folate and conjugated linolenic acid during fermentation and cold storage, enhancing the nutritional profile of the milk [70]. Probiotic LAB strains, such as *L. plantarum,* generate bioactive peptides with angiotensin-converting enzyme inhibitory and mineral-binding activities, contributing to cardiovascular health and mineral absorption [71]. Fermented dairy products exhibit hypocholesterolemic and antioxidant properties, which are beneficial for cardiovascular health [72]. The LAB strains in fermented milk can improve gut health by enhancing the intestinal microflora, reducing lactose intolerance, and preventing gastrointestinal infections [73]. Moreover, consumption of fermented dairy products can modulate the immune system and reduce allergic reactions [72]. LAB strains produce antimicrobial peptides and organic acids that inhibit pathogenic bacteria and fungi, enhancing the safety of fermented milk products [74]. Controlling post-acidification through strain selection and fermentation is crucial because LAB continue to produce lactic acid during storage, which can affect the flavor and shelf life of fermented milk [75]. Moreover, LAB strains such as *L. plantarum* and *L. delbrueckii* maintain high viability during storage, ensuring the continued probiotic benefits of the fermented milk [73].

#### 2.1.3. Yogurt Drinks

Although yogurt and yogurt drinks have been developed for centuries, their popularity and commercial modification are on a steady rise [76]. Yogurt is produced by fermenting milk with *L. delbrueckii* subsp. *bulgaricus* and *S. thermophilus*. These lactic acid bacteria generate lactic acid, which causes milk proteins to coagulate, resulting in the characteristic texture and flavor of yogurt [77]. Yogurt drinks are also known as drinkable yogurts, a popular type of dairy-based beverage that is consumed in liquid form. Their consistency ranges from runny to viscous with various flavors and sweetness. Ropy LAB strains (e.g., thermophilic culture YF-L81 composed of *S. thermophilus* and *L. delbrueckii* ssp. *bulgaricus*) produce exopolysaccharide that can improve the sensory, stability, and health benefits of the drinkable yogurts [78]. Enhanced with additional probiotics, prebiotics, and bioactive compounds, such as vitamins, antioxidants, minerals, and proteins, functional yogurt drinks offer extra health benefits for various conditions, such as constipation, diarrhea, colon cancer, irritable bowel syndrome, digestive-related allergies, *Helicobacter pylori* infection, elevated cholesterol and sugar levels, bone mineral loss, weight problems, and heart diseases. In addition, it can positively influence the psychology and mood of individuals, their behavior, and cognitive capabilities [79]. However, more studies are needed to enhance the stability of functional constituents and avoid any sensory changes during storage, as well as potential digestive discomfort for some individuals [77,79]. A type of drinkable yogurt produced in many countries in the Middle East is known as “laban”, which is available both commercially and in traditionally prepared in-house settings. It is simply made by diluting the fermented yogurt with water, adding salt, and mixing to release most of the fat, leaving the liquid portion as a drinkable yogurt. This drink is a part of the culture of many countries and considered a refreshing, nutritious liquid, especially in hot summer months. Microbiological analysis of the traditional laban in Oman revealed a dominance of mesophilic lactococci and homofemenentative lactobacilli, which included *L. lactis* ssp. *lactis*, *Lacrococcus locus* ssp. *locus*, *L. lactis* ssp. *cremoris*, *L. plantarum,* and, to a lower extent, the presence of *Leuconostoc* ssp. [80].

Fruit-flavored yogurts are made by adding fruit or fruit flavoring to enhance the sensory appeal and nutritional value of drinkable yogurts. Some researchers [81] formulated pomegranate and vanilla yogurt beverages that contained inulin as a prebiotic, along with the probiotic bacteria *Lactobacillus acidophilus* and *Bifidobacterium*, to obtain symbiotic products. Another set of samples was supplemented with approximately two volumes of carbon dioxide. The formulated beverages were stabilized with high-methoxyl pectin and whey protein concentrate and compared to the samples with added carbon dioxide. Both types of bacteria showed stability, demonstrated by bacterial levels greater than 10^6^ CFU/g in both flavors of beverage, both with and without carbonation. These carbonated symbiotic drinkable yogurts have potential for commercialization. Moreover, drinkable yogurts can be made with functional additives by incorporating functional ingredients, like coffee, tea, or plant extracts, that can improve the bioactive properties and offer unique flavors to the products [82,83,84]. For instance, black and green tea were demonstrated to enhance the total phenolic content and antioxidant activity of yogurt significantly [83]. Likewise, the addition of fig syrup was shown to enhance the sweetness and overall acceptability of drinkable yogurt [85]. These studies demonstrate a high potential to expand the variability of drinkable yogurts by incorporating various functional ingredients and commercializing them in the future.

### 2.2. Non-Dairy Fermented Drinks

#### 2.2.1. Vinegar

There are two steps in vinegar production. The first one is known as alcoholic fermentation, in which yeasts convert sugars to alcohol under anaerobic conditions. The second one is called acetous fermentation, in which acetic acid bacteria (AAB) convert the produced alcohol into acetic acid in the presence of oxygen [86,87,88]. Although LAB are not directly involved in these two key steps of vinegar production, their multiple roles cannot be ignored in vinegar production. For example, *L. plantarum* co-fermentation of mango vinegar was found to improve its flavor and taste by enhancing its content of vitamin C, total titratable acidity, volatile organic compounds, and other organic acids [89]. LAB, such as *Limosilactobacillus fermentum*, *L. plantarum*, *Lentilactobacillus buchneri*, *Lacticaseibacillus casei*, *L. lactis*, *Pediococcus acidilactici*, *Pediococcus pentosaceus*, and *Weissella confuse,* were reported as the natural microflora of vinegar [90]. Date vinegar is produced from dates, a fruit containing high amounts of the simple sugars—glucose and fructose [87,91]. Although various LAB were reported from different fruit vinegars, the LAB associated with date vinegar have rarely been analyzed. *Fructobacillus tropaeoli*, *Leuconostoc mesenteroides*, *Leuconostoc pseudomesenteroides*, and *Weissella paramesenteroides* were isolated from Persian date vinegar [90]. While our previous study [87] provided the levels of LAB in Omani date vinegar, their specific identities and characteristics remained undetermined.

In our previous work (unpublished data), we identified 27 LAB isolated from homemade and lab-made date vinegar samples using 16S rRNA gene sequencing [87]. All of them were Gram-positive bacilli, identified as *L. plantarum* (44%), *Levilactobacillus brevis* (30%), *L. paracasei* (15%), *L. rhamnosus* (7%), and *Companilactobacillus paralimentarius* (4%). These bacteria may play a significant role in fermentation and flavor development. *L. plantarum* is known to shape the microbial community and enhance flavor via volatile organic compound production [92,93,94]. *L. brevis* can contribute to a balanced microbial system via interaction with other microbes [95], contribute to the flavor and aroma [96,97], and tolerate high salt concentrations [93]. *L. paracasei* and *L. rhamnosus* are associated with flavor improvement, fermentation efficiency, and product safety [98,99,100]. Though rare in our isolates, *C. paralimentarius* has been reported in other fermented foods [101,102,103,104], suggesting a potential aromatic contribution.

#### 2.2.2. Wine and Beer

Many factors affect the quality of the complex beverage, wine, including the quality and genotype of the raw materials, the winemaking procedures, the aging vessels, and the microbes involved in the process, which include yeasts (particularly *Saccharomyces,* but sometimes non-*Saccharomyces* genus) that undergo alcoholic fermentation and the LAB responsible for malolactic fermentation (MLF), a secondary fermentation process that is crucial in winemaking [2,40]. The genera *Oenococcus*, *Lactobacillus*, *Leuconostoc*, and *Pediococcus* are the main LAB involved in winemaking. *Oenococcus oeni*, *Liquorilactobacillus mali*, *Liquorilactobacillus satsumensis*, and *L. paracasei* were used as starter cultures for MLF [105,106]. *O. oeni* has been often isolated from spontaneous MLF and is commonly used as a starter culture [105], although some studies reported that *L. plantarum* was the most efficient microbe in reducing the amounts of malic acid in the final product. *L. plantarum* can be inoculated before, during, or after the primary alcoholic fermentation process of fruit wine [89]. However, this bacterium showed a significant genetic variability among the same species, making their characterization for a particular regional wine important for refining the sensorial and other properties of wine [107]. The main role of LAB in MLF is to convert the dicarboxylic malic acid into monocarboxylic lactic acid and carbon dioxide, making the wine smoother and more palatable, as the concentration of malic acid that imparts tart flavor to wine is reduced. In addition, LAB’s enzymes, such as esterases, glycosidases, and proteases, have been proven to contribute to the production of many volatile organic compounds, which finally contribute to determining the texture, flavor, and appearance of the wine [2]. The MLF process, including bacteria such as *O. oeni* and *Pediococcus parvulus,* was also found to reduce the concentration of the mycotoxin ochratoxin A, which is produced in wine mainly by the molds *Aspergillus* and *Penicillium*. Mechanisms of cell-binding and biodegradation were proposed for such mycotoxin removal [40].

Recent advancements in alcoholic beverage fermentation have explored the utilization of non-traditional cultures, including non-Saccharomyces yeasts and LAB, to contribute to the unique flavors and improve the sensory characteristics of the final product [108]. Specific non-Saccharomyces yeasts like *Torulaspora delbrueckii* and *Candida* species have shown potential in enhancing the fermentation processes and the product quality [109]. The incorporation of probiotic strains and health-promoting compounds in alcoholic beverages is gaining traction. This approach aims to offer additional health benefits, such as improved gut health, alongside traditional alcoholic consumption. African traditional alcoholic beverages, for instance, are noted for their probiotic properties due to the active presence of LAB and yeasts during fermentation [110]. Moreover, advances in molecular biology and sequencing technologies are facilitating the identification and utilization of novel microbial strains. Techniques like whole-genome sequencing and third-generation DNA sequencing are being employed to better understand and harness microbial diversity [111].

Immobilized kefir culture is an innovative technique being explored in winemaking, particularly for producing low-alcohol wines. This method involves the physical confinement of kefir cells on various supports, which can enhance the fermentation efficiency and product quality [112]. For instance, low-alcohol wines (≤10.5% vol) were produced using wet and freeze-dried immobilized kefir cultures on natural supports. In comparison to the conventional free cell culture, the immobilized kefir culture showed high operational stability, and it was found suitable for simultaneous alcoholic and malolactic low-alcohol wine fermentation at temperatures up to 45 °C, with high ethanol productivity (up to 55.3 g/(Ld)) and malic acid conversion rates (up to 71.6%). The produced wine was considered high quality by the sensory panel, suggesting a potential industrial use in the semi-dry, low-alcohol wine-making at 37 °C and in producing novel wine products with a sweet (liquoreux) property at 45 °C, which is advantageous in regions with tropical climates or hot summer periods [113].

LAB can produce hydroxycinnamic acids from tartaric esters and decompose anthocyanin glucosides, affecting the color of wine. LAB help stabilize wine, thanks to their proteolytic action on digesting the proteins responsible for wine haze, thus lowering the need for bentonite addition. Because they exhibit pectinolytic activity, LAB can aid in wine clarification and the breakdown of acetaldehyde, including that bound to SO_2_, thus reducing the need for SO_2_ additions. The MLF can also reduce the concentration of the residual nutrients that can be used by spoilage microbes such as *Brettanomyces bruxellensis*, thereby improving the stability and shelf life of the final product [2,106]. LAB impart a buttery flavor to wine through the production of diacetyl from the fermentation of citrate [2,105].

On the other hand, some LAB species can negatively affect the quality of wine, leading to mousy taint, volatile acidity, bitterness, geranium notes, oily and slimy texture, and overt buttery properties. For example, *Pediococcus* sp. and *Lactobacillus* sp. can produce undesirable volatile phenols and N-heterocyclic and sulfur volatile compounds. Several procedures can be utilized to manage this spoilage, such as controlling wine acidity, adding sulfur dioxide, or using ultrahigh pressure, pulsed electric fields, and ultrasound or UV irradiation [40,114]. Moreover, LAB can produce compounds that are harmful for consumers, such as ethyl carbamate and biogenic amines, mainly histamine, tyramine, and putrescine. Bacteria such as *Enterococcus faecium*, *Lentilactobacillus hilgardii*, *L. brevis*, *L. plantarum*, *L. rhamnosus*, *O. oeni*, and *P. parvulus* were found to be involved in biogenic amine production. However, it is assumed that the ability to produce biogenic amines is strain-specific and not species-specific, and it depends on other factors such as the presence of nutrients and the decarboxylating activity of bacterial enzymes, which is favored in the low pH of wine [40].

LAB are integral to the production of traditional sour beer styles, such as Lambic and Gueuze, which rely on spontaneous fermentation. Modern methods use controlled fermentation with LAB to produce sour beers more efficiently. Brewers use sequential pitching of LAB and yeast, allowing optimal fermentation conditions for both microorganisms, leading to faster acidification and fermentation. LAB can lower the pH to around 3.5 within a short time by converting sugars into organic acids [115,116]. One study demonstrated the potential of using *L. brevis* BSO464 and *L. plantarum* in co-fermentation with the yeast *Saccharomyces cerevisiae* for controlled sour beer production with a shortened production time of 21 days [117]. The order of LAB pitching, before or after yeast, can significantly affect the fermentation outcomes, including the levels of lactic acid and the final pH of the beer. For instance, pitching LAB before yeast results in a higher lactic acid content and a lower pH, which is desirable for sour beers [116]. On the other hand, using mixed cultures of LAB and yeast can expedite the fermentation process and ensure consistency in flavor and quality. This method is faster than traditional spontaneous fermentation and allows for better control over the final product [117].

LAB can also be used to create probiotic beers, which offer health benefits in addition to unique flavors. These beers are formulated with probiotic LAB strains that survive the brewing process and remain viable in the final product. For instance, researchers produced probiotic-enriched Gueuze-style sour beer utilizing a two-step fermentation process that involved alcoholic fermentation utilizing *Saccharomyces boulardii* CNCM 1-745 and lactic acid fermentation utilizing *L. acidophilus* LA3, *L. acidophilus* LA5, *L. plantarum* 299v, *L. rhamnosus* GG, and *L. pseudomesenteroides* BIOTEC011. LAB fermentation enhanced the content of organic acids and volatile compounds, as well as the sensory characteristics. Moreover, high levels of probiotics were retained under different conditions, such as carbonation, storage at 4 °C, and simulated gastrointestinal digestion [118]. A novel type of beer was produced using olive leaves as an ingredient. Olive leaves significantly increased the polyphenol content of beers, while their addition did not influence the antioxidant activity. About 5 g/L of olive leaves resulted in a beer with a pleasant sensory profile [119]. This kind of research opens the door for utilizing new ingredients for beer manufacturing, particularly adding a nutraceutical value.

However, while LAB are beneficial in controlled fermentations, they are also the most common spoilage bacteria in beer, capable of producing off-flavors (lactic acid, diacetyl) and turbidity, rendering the beer undrinkable. *L. brevis* was identified as a common spoiler in bottled microbrewed beer from Australia [120]. Other LAB, such as *Liquorilactobacillus acetotolerans*, *L. plantarum*, and *Pediococcus damnosus,* were described as beer-spoiling bacteria [121]. Therefore, effective management of LAB strains and brewing conditions is essential to prevent spoilage. LAB used in beer production must be hop-tolerant, as hop acids can inhibit many bacterial strains. Specific LAB strains, such as *L. plantarum* and *L. brevis*, have been identified for their hop tolerance and suitability for beer fermentation [117,122].

#### 2.2.3. Plant-Based Fermented Juices

Produced through the action of yeasts, LAB, and acetic acid bacteria (AAB) and loaded with bioactive compounds, fruit-based/plant-based fermented juices have become one of the consumers’ preferred beverages recently. An example is water kefir, which is a non-dairy functional beverage made from water, sugar, and dried fruit fermented with the gelatinous kefir grains composed of a symbiotic microbial consortium surrounded with exopolysaccharide, produced mainly by LAB, while the yeasts provide the nitrogen source needed for the microbial assimilation reactions. This symbiosis makes kefir grains remain stable for years. The grains are translucent, colored gray–white, waxy, and tough [49,55,123,124]. The exact microbial strains found in the kefir grain vary greatly according to the geographical origin of the grains and their substrate and conditions of fermentation, and this has tremendous effects on the sensorial characteristics of the beverages; however, manipulating these factors can refine and customize the final product. The grains of water kefir and milk kefir are not interchangeable, as the former grains require vegetable-, fruit-, or cereal-based liquids with adequate levels of fermentable fructose or sucrose [55]. Various LAB genera were identified in water kefir, including *Lactobacillus*, *Leuconostoc*, and *Streptococcus*. However, *Zymomonas mobilis,* which is a non-LAB, non-AAB gluconic acid producer, was found to be a dominant bacterium in kefir, in which it significantly contributes to the flavor of water kefir, besides *Bifidobacterium*, *Acetobacter*, *Saccharomyces*, and *Lactoplantibacillus* genera [49,123,124]. Water kefir is assumed to deliver many benefits to the body through modulating gut microbiota, enhancing immune system functions, increasing nutrient absorption, and improving metabolic potential [125].

Kombucha is another fermented beverage made from black tea with added sugar and fermented by the action of yeasts, LAB, and AAB. In this case, the consortium is found in a cellulosic mat, and the yeasts such as *Zygosaccharomyces*, *Candida,* and *Lachancea* dominate the fermentation process, followed by AAB. In addition, LAB, such as *Lactobacillus*, *Leuconostoc,* and *Lactococcus,* are present from the beginning of the fermentation process and increase in number as the fermentation proceeds, reaching up to 10–30% of the microbiota by the end of the fermentation process [94]. The water extract of soybeans is also fermented to produce soymilk by the action of the LAB species *L. acidophilus*, *L. bulgaricus*, *L. casei*, and *S. thermophilus*. This probiotic product is rich in antioxidants and used in the treatment of various health conditions, such as breast cancer and gut problems [126].

The fermentation of cabbage by LAB, such as *L. brevis*, *L. plantarum*, *L. mesenteroides*, and *Lactobacillus sakei*, leads to a product known as sauerkraut that also contains juice consumed in some regions in the Black Sea as a beverage. Recently, attempts have been made to ferment various fruits and vegetables using LAB, such as *Lactobacillus* spp., *Leuconostoc* spp., *Weissella* spp., or water kefir and applying mild pasteurization treatment to the raw material to inhibit undesirable microbes and permit better conditions for LAB fermentation [49]. The LAB strains (*L. plantarum* and *L. fermentum*) showed promising potential for fruit juices, such as blueberry juice fermentation. Within 48 h of fermentation, these bacteria were able to increase the lactic acid concentration, reduce the malic acid concentration, increase the phenolic content to over 80%, and enhance the antioxidant potential by 34.0%. In addition, the anthocyanin content decreased with fermentation, while the levels of p-hydroxybenzoic acid and caffeic acid declined [127]. Likewise, another study [128] that characterized “summer black” grape juice fermented by the strains of LAB (*L. rhamnosus*, *L. paracasei* subsp. *paracasei*, *L. delbrueckii* subsp. *bulgaricus*, *L. plantarum*, and *S. thermophilus*) reported a significant improvement in the content of tartaric acid, malic acid, and citric acid. Fruit and floral aroma compounds, such as phenylethyl alcohol, lactic ethyl ester, L-menthol, citronellol, and nerol, were some of the products of the fermentation process. A mixture of wheat bran with root vegetables (red beetroot and carrots) was also utilized as a fermentation substrate for *L. plantarum* BR9, *L. plantarum* P35, and *L. acidophilus* IBB801. The obtained beverages combined the nutritiveness of the raw materials with the beneficial effects of the fermentation process, such as the enhancement of flavor, aroma, shelf life, safety of products, and their health benefits [14].

The production of fermented legume beverages using LAB is a growing area of interest due to the health benefits and enhanced nutritional properties these beverages offer. The fermentation process typically involves mixing legumes with water and other ingredients, inoculating with LAB, and fermenting under controlled conditions (e.g., specific temperatures and pH levels). For instance, a study on wheat germ and sweet-waxy maize used a combination of *L. bulgaricus*, *S. thermophilus*, and *Bifidobacterium lactis* for fermentation at 38 °C and pH 7.0 for 24 h. The produced beverage tasted slightly sweet and sour and had the refreshing flavor of wheat germ and sweet-waxy maize [129]. Additional processing steps, such as heat treatment and the addition of stabilizers, may be required to ensure the long-term storage and consistent quality of the fermented legume-based beverages [49,129]. LAB fermentation improves the nutritional profile of legume-based beverages by increasing the bioavailability of essential nutrients, such as amino acids, minerals, and vitamins. It also leads to the accumulation of bioactive compounds, like exopolysaccharides, short-chain fatty acids, and bioactive peptides, which contribute to the health benefits of these beverages [130]. Moreover, LAB fermentation effectively reduces the antinutritional factors in legumes, such as trypsin inhibitors, cyanide, saponins, raffinose series oligosaccharides, tannins, and phytates. This reduction enhances the digestibility and nutritional value of the legumes [131]. Fermented legume beverages can serve as synbiotic foods, combining probiotics (beneficial bacteria) and prebiotics (compounds that support the growth of beneficial bacteria). This is because legumes naturally contain prebiotic ingredients like oligosaccharides, resistant starch, polyphenols, and isoflavones, which support the growth of LAB [130].

Various LAB strains are used in the fermentation of legume beverages, including *L. plantarum*, *L. lactis*, and other *Lactobacillus* species. These strains are selected for their ability to produce organic acids, antimicrobial substances, and other beneficial metabolites [132]. A study found that *L. pseudomesenteroides* significantly catabolized raffinose, maltose, and citrate, which are present in soy beverages, while *L. lactis* produced high concentrations of diacetyl and lactic acid, which are relevant for the generation of plant dairy alternatives. It also decomposed phytic acid, pectin, and sucrose, mostly present in bean, cereal, and fruit-based plant matrices [133]. Likewise, the fermentation of an Apulian black chickpea protein concentrate using *S. thermophilus* alone and its co-cultures, *L. lactis* and *L. plantarum,* yielded beverages with high protein (120.00 g/kg) and low fat (17.12 g/kg) contents, while the levels of phytic acid decreased and saturated fatty acids largely decreased. The formulated beverages had greater lightness, consistency, cohesivity, and viscosity than the controls. Interestingly, the aromas of legumes and grass were not evident in these beverages, possibly due to the formation of new volatile organic molecules [134]. In another study, researchers fermented the water extracts of lupin, pea, and bean grains by inoculating them with *L. acidophilus* ATCC 4356, *L. fermentum* DSM 20052, and *L. paracasei* subsp. *paracasei* DSM 20312. The fermentation of bean water extract resulted in an unpleasant ferric-sulfurous off-odor. However, lupin- and pea legume-based beverages had improved sensory characteristics and retained high levels of viable LAB until the end of the cold storage [135].

Cereal-based beverages fermented with LAB have gained popularity due to their functional and nutritional benefits. The production process typically involves the fermentation of cereal substrates by LAB, which can be done using monocultures or co-cultures with other microorganisms, such as yeasts. Common cereals used include malt, rice, maize, barley, and buckwheat. LAB such as *L. rhamnosus*, *L. acidophilus*, *L. plantarum*, and *Lactobacillus helveticus* are frequently used. These bacteria can be used alone or in combination with yeasts, like *S. cerevisiae* [136,137,138,139,140]. Enzymes like α-amylase, protease, and glucoamylase are often used to hydrolyze the cereal substrates, making sugars and amino nitrogen more available for bacterial growth [137,138]. Moreover, parameters such as temperature, pH, and fermentation time are optimized to enhance the bacterial growth and product quality [138]. The fermentation process enhances the functional potential of cereal-based beverages. Fermented cereal beverages are rich in organic acids, free amino acids, and bioactive compounds like gamma-aminobutyric acid [136,137,138]. These beverages often contain high counts of viable LAB, which can confer probiotic benefits, including improved gut health and enhanced immune function [139]. In addition, the fermentation process improves the flavor, aroma, and texture of the beverages. LAB contribute to the development of desirable sensory attributes by producing volatile compounds and organic acids [137,138,140,141,142]. Similar to other LAB-fermented beverages, the acidic environment created by LAB fermentation inhibits the growth of pathogenic bacteria, thereby enhancing the safety and shelf life of the beverages [142]. Multi-cereal beverages can also be produced by combining different cereals, like malt, rice, and maize, fermented with LAB and yeasts, yielding beverages with unique flavors and high nutritional value [138]. Thus, cereal-based beverages fermented with LAB offer a promising avenue for developing functional foods with enhanced nutritional and sensory properties. The use of specific LAB strains and optimization of fermentation conditions are crucial for producing high-quality beverages that meet consumer demands for health and taste [13,136,137,138].

### 2.3. Hybrid Fermented Beverages

The first hybrid dairy product introduced in the market in 2021 was an alternative drink made from a blend of equal proportions of cow’s milk and plant-based ingredients (soy, oat, or almonds) [143]. Following this, researchers have taken it further to produce hybrid fermented drinks made from dairy and non-dairy raw materials. In one study [7], an attempt was made to produce a fermented beverage from milk–oat that was prepared by mixing cow’s milk and oat beverage in a 1:1 ratio and inoculating it with the LAB *S. thermophilus, L. delbrueckii* subsp. *Bulgaricus,* and/or *L. plantarum* 299v and *L. acidophilus* La5. The bacterial mixtures and the type of raw material influenced the final product’s sensory attributes. The authors noted that more research is needed to further characterize these new types of dairy and non-dairy hybrid beverages, as they can offer nutritional benefits of both types of raw materials with enhanced sensory characteristics. Likewise, the fermentation of a mixture of cow’s milk and soybean milk using the probiotic bacteria *L. acidophilus* La5 was demonstrated to enhance the sensory properties of the beverage product with good acceptance by potential consumers [144]. These studies indicate that the LAB’s potential to transform a wide range of raw materials into new products that meet consumers’ expanding demands is unlimited, providing the agricultural, industrial, and health sectors with an open renewable source fueled by the LAB’s versatile and functional genetic reservoir [145,146,147,148,149].

Hybrid yogurt was produced by blending cow’s milk with soy and oat drinks in various ratios. The hybrid yogurt resulted in improved viscosity, a favorable pH gradient, and the absence of pathogens in the final product, demonstrating the microbial safety of the products. This approach in hybrid yogurt production can enhance consumer acceptance by combining dairy and plant-based derivatives [150]. Hybrid fermented beverages with LAB are rich in various bioactive compounds produced during fermentation. For instance, a fermented milk product was prepared by mixing cow’s milk and quinoa beverage with a starter culture containing *L. acidophilus* and *Bifidobacterium bifidum*. The addition of quinoa beverage stimulated the growth of the starter culture and yielded a final product with higher total phenolic content, minerals, antioxidant activity, and amino acids than the control [143]. An attempt was also made to produce kefir via the addition of quinoa flour or rice flour to cow’s milk. The fortification of kefir with quinoa flour reduced the fermentation time by 2.5 h, while the fortification with rice flour reduced it by 1.5 h in comparison to the control. However, the controls received better acceptance by the panelists, probably due to their better acidity. The kefir fortified with 0.5% quinoa obtained the highest viscosity and acidity. The reduced fermentation time was due to the addition of prebiotics from quinoa and rice flours that are sources of proteins and polysaccharides [151].

A new yogurt was prepared from cow’s milk blended with watermelon seed milk and inoculated with *Streptococcus salivarius* subsp. *thermophilus* EMCC104 and *L. delbruekii* subsp. *bulgaricus* EMCC1102. Hyperuricemic rats fed a diet supplemented with 10% watermelon seed–milk yogurt showed a significant improvement in renal function compared to the control group. This effect could be due to the increased antioxidant activity via enhancement of the functions of superoxide dismutase, catalase, and glutathione transferase. This example shows that byproducts of food waste can be combined with dairy materials to produce novel hybrid beverages with enhanced functional and nutritional properties [152]. Another study produced a functional yogurt drink fortified with golden berry juice and examined its therapeutic effect on rats with hepatitis. The yogurt drinks fortified with golden berry juice had the highest content of total phenolic compounds, antioxidant activity, and organoleptic scores compared to the controls. Rats fed on a diet fortified with functional yogurt drinks containing golden berry juice for 8 weeks showed higher potential hepatoprotective effects compared with the liver injury control group, highlighting the potential of using this hybrid drink in protecting the liver [153].

A summary of some examples of the different types of beverages produced by LAB fermentation is presented in Table 2.

## 3. Metabolic Activities of LAB During Beverage Fermentation

The metabolic activity of LAB determines their role in the sensory profile, quality, safety, shelf life, and health benefits of fermented foods. Various bioactive compounds can be produced by LAB’s unique metabolic systems, as summarized in Table 3. Environmental conditions influence the metabolism of LAB, and these conditions can be manipulated to achieve more favorable fermentation outcomes [154]. LAB contribute to the flavor of fermented foods through three main pathways, each utilizing different substrates: the fermentation of sugars (glycolysis), the degradation of proteins (proteolysis), and the degradation of lipids (lipolysis). Protein degradation can produce peptides and free amino acids, which can further be converted into aldehydes, alcohols, acids, esters, and sulfur compounds [3].

Based on their glucose fermentation pathway and end products, LAB can be classified into obligate homofermentative, facultative heterofermentative, or obligate heterofermentative [154]. A comparison between LAB homofermentation and heterofermentation is presented in Table 4.

### 3.1. Carbohydrate Metabolism

LAB metabolize carbohydrates primarily through homofermentative (producing lactic acid) and heterofermentative (producing lactic acid, ethanol, acetate, and CO_2_) pathways. This activity is essential for the production of lactic acid, which lowers the pH and acts as a preservative in beverages and other fermented foods [174,175]. LAB homofermentative metabolism occurs through the Embden–Meyerhof–Parnas pathway, converting glucose into two molecules of pyruvate, which is then reduced to lactic acid via the enzyme lactate dehydrogenase, yielding two molecules of ATP (adenosine triphosphate) per glucose, with no production of CO_2_ [3]. In this pathway, the fate of pyruvate is determined by the type of substrate and the presence of oxygen. Pyruvate formate lyase (PFL), an enzyme that can convert pyruvate into formate and acetyl-CoA, operates under anaerobic conditions, particularly during limited substrate availability. Bacterial starter cultures using this pathway are best implemented when lactic acid production is desirable [154].

Heterofermentative metabolism of carbohydrates occurs via the phosphoketolase pathway in LAB, yielding one ATP molecule per glucose, lactic acid, and other byproducts, such as ethanol, acetic acid, and CO_2_ [176,177]. This pathway is particularly desirable in fermentation processes when complex flavors and gas are beneficial [3]. Lactate remains the main product of the heterolactic metabolism, but acetate production contributes to both energy yield and flavor/aroma development. Acetate, while beneficial for its sour taste and antimicrobial properties, can be detrimental in high concentrations, particularly in alcoholic beverages, leading to spoilage. Heterofermentative LAB are present in most plant-based fermented products, such as vinegar, wine, cereal, sauerkraut, sourdough, and kimchi, reflecting the availability of substrates, such as sucrose, maltose, raffinose, and other electron acceptors [154]. Moreover, LAB can ferment various sugars, including glucose, fructose, and sucrose, but they generally do not ferment polysaccharides like starch due to the lack of necessary hydrolytic enzymes [32]. This sugar fermentation is critical for energy production and the generation of metabolic by-products that influence the flavor and texture of the beverage [178].

### 3.2. Organic Acid Metabolism

In beverages like wine and cider, LAB convert L-malic acid to L-lactic acid (malolactic fermentation; MLF), reducing acidity and enhancing flavor complexity. This process can also produce other organic acids and ethanol, impacting the sensory qualities of the final product [175,179]. Previous results showed that different strains of LAB that have been collected from different environments possess different amino acid decarboxylases, highlighting the complex metabolic systems in LAB [180]. A study showed that almost all homofermentative LAB isolated from commercial fermentations were able to undergo malate decarboxylation, whereas half or fewer heterofermentative LAB were able to do so. *L. mesenteroides* was demonstrated to carry out the malolactic reaction in cabbage juice, indicating possible control of the fermentation process [181]. In addition, LAB metabolize pyruvate and lactate, which are central to their energy production, and influence the organoleptic properties of fermented beverages [174].

### 3.3. Amino Acid and Protein Metabolism

LAB can convert amino acids into various compounds, including biogenic amines, which can have both beneficial and harmful effects on the beverage’s quality [182]. This conversion is part of their broader metabolic activities that contribute to the flavor profile of the beverage. The catabolism of amino acids begins with a transamination reaction, leading to the formation of α-keto acids, which can be decarboxylated to produce aldehydes, contributing to flavor development [183,184]. Glutamate dehydrogenase (GDH) is involved in the direct production of α-ketoglutarate, a molecule essential for amino acid transamination [184,185]. *L. lactis* was demonstrated to produce diacetyl and acetoin as a result of aspartate or alanine catabolism in the presence of α-ketoglutarate culture medium [186]. Among the flavor compounds, branched-chain amino acids are converted into compounds contributing to fruity, malty, and sweaty flavors. The catabolism of aromatic amino acids produces floral chemical flavors. The catabolism of aspartic acid produces compounds with buttery flavor, while sulfur-containing amino acids are converted into compounds with garlic, boiled cabbage, or meaty flavors [187].

### 3.4. Polyunsaturated Fatty Acid (PUFA) Metabolism

LAB utilize a complex enzyme system to metabolize PUFAs to produce unique bioactive fatty acids, such as hydroxy fatty acids and conjugated fatty acids (Table 3). Hydratase, dehydrogenase, isomerase, and enone reductase are the four key enzymes involved in accomplishing the saturation of a C=C double bond and generating unique bioactive fatty acids (Table 3), such as hydroxy fatty acids, conjugated fatty acids, and oxo fatty acids [188,189]. When cow’s milk and soy beverages were supplemented with flaxseed extracts, which are rich in PUFAs, LAB enhanced the nutritional properties by producing bioactive metabolites. These include enterolactone, matairesinol, and various flavonoids, which are beneficial to human health [190]. Moreover, the metabolic activities of LAB not only modify the fatty acid composition but also influence the sensory qualities of fermented beverages. The production of specific fatty acids and their derivatives can enhance the flavor, aroma, and overall quality of the beverages [191].

## 4. Methods for Isolation and Identification of LAB from Beverages

Following the sample collection of various fermented beverages, LAB can be isolated on selective media, such as de Man, Rogosa and Sharpe (MRS) agar, which is commonly used for LAB isolation due to its nutrient composition. Some LAB strains are difficult to isolate and grow on artificial media due to their complex nutritional requirements, low pH or oxygen sensitivity, and their slow growth rates. Enrichment techniques, such as supplementing MRS broth with ethanol (5%) and 15% (*v*/*v*) tomato juice and autoenrichment in grape juice homogenate, were utilized to culture LAB from wine grapes, which were also compared to using MRS without supplements. The genera *Lactobacillus*, *Enterococcus*, *Lactococcus,* and *Weissella* were identified by polymerase chain reaction (PCR) and PCR–denaturing gradient gel electrophoresis (DGGE). *Fructilactobacillus Lindneri*, *Apilactobacillus Kunkeei*, *L. lactis*, and *L. Plantarum* were recovered using all types of enrichment methods or MRS alone. *L. Kefiri* was detected only from MRS without enrichment, while *L. mali* was detected from all other enrichment types, but not from MRS alone [192]. This emphasizes the importance of utilizing diverse enrichment techniques and, possibly, culturing media and conditions (temperature, O_2_ and CO_2_ levels, pH) when isolating LAB from the same source [193,194].

Enrichment cultures in media supplemented with specific carbohydrates (e.g., fructose, mannitol) were found to enhance the isolation of diverse LAB species from the fermented products. For instance, to isolate autochthonous LAB from traditional fermented beverages in southern Africa (amasi, a fermented milk beverage; mahewu, a cereal-based non-alcoholic beverage; and tshwala, an alcoholic beverage), the media was enriched with 13 different carbohydrates. These carbohydrates had a huge impact on the diversity and numbers of the isolated LAB, in which fructose and mannitol were selected for the highest LAB numbers compared to other carbohydrates. Interestingly, the same study demonstrated little influence of the type of sugar in culture media on LAB isolation from fresh plant samples, such as fruits and flowers. This may indicate that LAB in these fermented products have adapted to utilize certain sugars efficiently [195].

A combination of conventional and molecular techniques can be utilized for the accurate and reliable identification of LAB from beverages and other foods [196,197,198]. The advantages and limitations of various methods used for LAB identification are summarized in Table 5. The initial identification of LAB can involve phenotypic testing, such as Gram staining, cell shape and arrangement, catalase and oxidase testing, and motility testing [199]. Biochemical tests such as API50CHL can be utilized to determine the carbohydrate fermentation profiles [200]. The conventional methods have various limitations. Phenotypic methods can sometimes misidentify LAB due to the overlapping characteristics among different species [201]. These methods often require extensive sample preparation and can be time-consuming, making them less efficient for rapid identification needs. Conventional phenotypic methods often involve destructive testing, which can limit the ability to perform further analyses on the same sample [202]. Moreover, the phenotypic methods may not be sensitive enough to detect LAB in mixed cultures or under stressful conditions, where bacteria may enter a viable but non-culturable (VBNC) state [203].

The automated machines, such as VITEK 2 compact, can be utilized for the identification of some LAB, while other more advanced machines, such as MALDI-TOF MS (matrix-assisted laser desorption/ionization time-of-flight mass spectrometry), can provide the identification of more LAB species, depending on the coverage of their databases. MALDI-TOF MS can quickly identify bacteria by analyzing their protein fingerprints, making it suitable for high-throughput applications. The method involves constructing extensive identification databases and using peak-based numerical analysis to achieve species-level identification. MALDI-TOF MS has been successfully used to identify beer spoilage bacteria, such as *L. brevis* and *P. damnosus*. This is crucial for quality control in breweries to prevent microbial outbreaks that can lead to product recalls. Challenges in using MALDI-TOF MS include identifying bacteria in mixed cultures due to compatibility issues. However, novel approaches combining inertial microfluidics and secondary flows have been proposed to separate and identify spoilage microorganisms from mixed cultures efficiently [208,209]. In addition, the accuracy of MALDI-TOF MS depends on the comprehensiveness of the reference database. Additional reference strains may be necessary to increase the sensitivity and specificity for species-level identification [210].

The 16S rRNA gene sequencing is a widely used method for LAB identification at the species level. Universal primers are utilized to amplify the gene, after which it is sequenced, and the sequences are compared to known sequences in databases [194,196,211]. Sequencing other genes, such as the 16S-23S rRNA gene intergenic spacer region, *pheS*, *rpoA*, and *recA* genes, can help discriminate between LAB species [212,213]. Whole genome sequencing (WGS) can also provide a comprehensive identification and characterization of LAB, including insights into their genetic diversity and evolutionary relationships [194]. Utilizing WGS and mining, novel biosynthetic gene clusters were identified in *L. plantarum* TXZ2-35, *L. fermentum* TZ-22, and *Companilactobacillus crustorum* QHS-4 isolated from traditional fermented milk. The bacteria were proven to inhibit *L. monocytogenes* in cheese, improving its safety as well as its sensory quality [214].

Next-generation sequencing (NGS) methods can provide detailed insights into the LAB microbial diversity and dynamics during fermentation processes. Techniques such as DGGE and TRFLP (terminal restriction fragment length polymorphism) can be utilized to identify the LAB present in mixed samples without their isolation [192,207]. These culture-independent molecular fingerprinting techniques are good for comparing microbial community structure between different samples or treatments. Moreover, metagenomics is a powerful tool to study LAB due to its ability to analyze genetic material recovered directly from the food samples, which is very typical for LAB performing fermentation in beverages, providing insights into their diversity, functionality, and interactions within complex microbial communities [214,215,216]. High-throughput sequencing (HTS) of the 16S rRNA gene can also be used to analyze the diversity and dynamics of LAB during fermentation processes. Some researchers analyzed the diversity and dynamics of LAB in atole agrio, a traditional maize-based beverage of Mexican origin. The HTS of the 16S rRNA gene confirmed the predominance of *Lactobacillaceae* and *Leuconostocaceae* at the beginning of the fermentation process. This technique also revealed that *L. plantarum* predominated in the liquid batches, whereas *W. confusa* was the dominant species in the solid batches [217]. Many LAB are identified and examined to test their potential use as probiotics or starter cultures. In one study [218], six LAB strains (*L. fermentum* 73B, *P. pentosaceus* 74D, *L. fermentum* 44B, *W. confusa* 44D, *L. fermentum* 82C, and *Weissella* cibaria 83E) that were isolated from spontaneously fermented Ethiopian cereal-based beverages (naaqe and cheka) were demonstrated to have antipathogenic, immunostimulatory, and starter culture potentials. The authors concluded that these bacteria can be used as autochthonous probiotic starters for naaqe and cheka fermentations after confirming their safety. Other high-throughput sequencing technologies, such as PacBio SMRT and Illumina MiSeq, are used to analyze the entire microbial community in fermented beverages. These methods can identify a wide range of LAB species and their relative abundances [219,220].

## 5. Challenges and Future Perspectives in Using LAB in Beverages

Using LAB to ferment a wide variety of liquid-based foods containing either single or mixed food types can open the door to an unlimited number of novel beverages with enhanced functional properties and nutritional value [13]. However, there are still challenges in optimizing the process to obtain products with desirable properties concerning sensory attributes, bioactive compounds, and longer shelf life [221]. Moreover, the production of LAB-fermented beverages can provide means for promoting the circular economy through the utilization of agro-industrial wastes and non-conventional sources, such as fruit pulps and peels, potato peels, and faba beans [222]. Byproducts from the beverage industry, such as spent grains from brewing and pomace from winemaking, are also utilized. These contain valuable bioactive compounds, like polyphenols and dietary fibers, that can enhance the functionality of the fermented beverages [223]. For instance, watermelon seed milk was utilized in yogurt manufacturing and was demonstrated to improve kidney function as an anti-hyperuricemic agent [152]. LAB have various adaptation mechanisms that permit them to utilize various carbon sources, including sugars and organic acids, which are abundant in food by-products and waste extracts. For instance, LAB convert L-malic acid to L-lactic acid during malolactic fermentation in wines and ciders, enhancing flavor and stability. They can also metabolize glycosides and other carbon sources, contributing to the sensory quality of the fermented products. LAB can thrive in various environmental conditions, including different pH levels and temperatures. For instance, *L. plantarum* can survive at high cell densities in orange juice at 4 °C without altering its organoleptic properties, making it suitable for functional juice production [224]. On the genetic level, LAB have undergone gene loss and gain events, allowing them to adapt to nutrient-rich niches and dominate specific habitats. This genomic plasticity is crucial for their survival and efficiency in diverse fermentation environments [225,226].

The selection of suitable strains of LAB for a particular food fermentation is essential for attaining the desirable outcomes. Different strains respond differently to different environmental conditions. For example, media sterilization was found to increase the cell concentrations and decrease the cell volumes of five strains of *L. delbrueckii* [227]. Moreover, the fermentation process should be optimized to maximize the beneficial effects of LAB, maintaining product consistency at the same time; factors such as pH, temperature, and nutrient availability should be managed [13,115]. In addition, if attempts have been made to produce novel beverages, their safety should be proved and approved by regulatory agencies. This is because even though most LAB strains are “generally recognized as safe” (GRAS) by the US FDA (Food and Drug Administration) and as having Qualified Presumption of Safety (QPS) by the European Food Safety Authority [228], using them in new products can lead to the production of substances for which safety should be checked [229,230]. Genetically modified LAB (GM-LAB) face stricter regulatory scrutiny compared to traditional LAB strains. This is primarily due to concerns about the dissemination of antibiotic resistance genes and the environmental impact of GM organisms [230]. One of the major concerns with GM-LAB is the potential for the horizontal gene transfer of antibiotic resistance markers, which can pose significant public health risks. Regulatory bodies require thorough evaluation of antibiotic resistance profiles and the potential for gene transfer before approving GM-LAB for use [231]. To mitigate risks, strategies such as biocontainment, use of food-grade selection markers, and homologous DNA are recommended. These measures aim to prevent the release of GM-LAB into the environment and ensure their safe use in closed systems [230]. Finally, consumer acceptance of any novel product will be necessary for its success in the market; therefore, educating them about the health benefits can help them in accepting these products [232,233].

In the future, LAB can be exploited more for heavy metal and other toxic compound remediation. Heavy metals, such as arsenic (As), aluminum (Al), cadmium (Cd), cobalt (Co), copper (Cu), chromium (Cr), lead (Pb), manganese (Mg), mercury (Hg), nickel (Ni), and vanadium (V), are toxic to the body. Their contamination of soil, water, and food and accumulation in the body has become a significant health concern globally. The main route for the entry of heavy metals into the body is through food and water. Heavy metals can alter the function and composition of gut microbiota, such as increasing the ratio of Bacteroidetes-to-Firmicutes ratio. This dysbiosis leads to increased lipopolysaccharide production and reduced microbial metabolism of short-chain fatty acids. Heavy metals can also damage the intestinal epithelial cells, leading to a leaky gut and the production of harmful metabolites, contributing to inflammation, allergies, cardiovascular diseases, diabetes, and cancer. LAB can be utilized for heavy metal removal in water, beverages, food, and soil [234]. LAB-fermented foods and beverages were demonstrated to help detoxify heavy metals in the tissues and blood of the human body [235,236].

The main mechanisms that LAB utilize for bioremediation are adsorption and bioaccumulation. The adsorption mechanism can happen both inside and outside the cells. Proteins and sugars present as extracellular polymers on the cell surface and metal-binding proteins inside the cell can chelate with metal ions to stick them on the cell. Then, LAB utilizes oxidation–reduction, methylation, and other reactions as means for the biotransformation that takes place inside the cell, decreasing the toxicity of these metals. However, the efficiency of LAB remediation of heavy metals is not high, highlighting the need for more research to improve the adsorption process, probably through pretreatment and the utilization of mixed LAB cultures. Compared to other microbes, LAB can grow strongly and can adapt to adverse environmental conditions. As probiotics and generally regarded as safe microorganisms, LAB can be utilized for the bioremediation of heavy metals in food and water. Mixed LAB cultures containing *E. faecium*, *L. plantarum*, and *L. fermentum* showed a synergistic effect in heavy metal bioremediation as compared to single bacterial cultures. In the future, techniques such as genetic engineering, bacterial immobilization, and the nanomaterials combination can be utilized to improve adsorption to heavy metals, reduce the process cost, increase strain stability and functionality, and enhance sustainability and environmental impact [234].

Enhancing LAB tolerance to acid through adaptive evolution was also studied to develop LAB strains capable of withstanding harsh acid stress conditions. Investigators challenged wild-type *L. mesenteroides* with increased concentrations of lactic acid for one year. One of the three mutants (LMS50, LMS60, and LMS70), LMS70, doubled D-lactic acid production compared to the wild type. These experiments proved genetic and physiological plasticity in LAB as beneficial mutations increased, leading to increased production of D-lactic acid [237]. In addition, in situ or in-stream product recovery (ISPR) techniques for lactic acid removal during fermentation help reduce product inhibition and enhance overall process efficiency [238]. Moreover, co-cultivating LAB with other microbes can increase lactic acid production and lower production costs. Co-cultivating *L. lactis* with four cellulose-synthesizing bacteria increased the production of both bacterial cellulose and lactic acid, reducing production costs and enhancing sustainable biomaterial production [239]. Microwave-assisted deep eutectic solvent pretreatment has also been proposed as a solution, reducing the pretreatment time to less than a minute. Deep eutectic solvents contain two types of compounds: hydrogen bond donors (e.g., polyols, carboxylic acid, amides, and amines) and hydrogen bond acceptors (e.g., quaternary ammonium salts and choline chloride). They are preferred for the pretreatment of the feedstock because they are cheap, biodegradable, easy to prepare and recycle, and produce no or low toxic by-products [238].

LAB have been genetically manipulated to produce functional molecules in vitro or in vivo. The examples include site-specific chromosomal deletions, mutations, stable integrations, insertions, introducing DNA into LAB cells, and genome editing of LAB using the clustered regularly interspaced short palindromic repeats–CRISPR-associated proteins (CRISPR–Cas) system [240]. These techniques are promising tools for LAB genetic engineering for efficient utilization in various sectors, including foods and beverages, to optimize fermentation outcomes. Moreover, LAB have emerged as promising vehicles for drug delivery due to their safety, probiotic features, and ability to survive in the gastrointestinal tract [241,242,243]. LAB and other probiotic bacteria can be used to deliver therapeutic molecules directly to the mucosal tissue [244], for example, anti-inflammatory compounds to treat inflammatory bowel disease [245,246]. LAB have been utilized as live vectors for mucosal vaccines, expressing antigens to provoke immune responses against various pathogens [247,248]. Coupling LAB involved in beverage and other food fermentation with their utilization as drug delivery systems can enhance the functionality of foods, reduce drug production costs, and enhance the stability and bioavailability of drugs [243].

## 6. Conclusions

LAB have different metabolic pathways, such as carbohydrate metabolism, organic acid metabolism, protein metabolism, and polyunsaturated fatty acid (PUFA) metabolism. LAB fermentation enhances the flavor profile of beverages by producing organic acids, such as lactic acid, that contribute to a tangy taste. The type of LAB and substrate used can significantly impact the texture and stability of the fermented beverages. LAB can metabolize phenolic acids, resulting in bioactive metabolites, which suppress pro-inflammatory cytokines. LAB fermentation increases the antioxidant activity of beverages, enhancing their health-promoting properties. Bacteriocins and organic acids produced by LAB can inhibit pathogenic microorganisms, contributing to food safety and health benefits. LAB fermentation can enrich beverages with bioactive compounds, like γ-amino butyric acid (GABA), which has various health benefits such as antihypertensive effects, neuroprotective and cognitive benefits, metabolic and endocrine benefits, and mental health benefits. The proven health benefits of fermented beverages, including dairy, non-dairy, and hybrid beverages, have stimulated and attracted more consumers toward this sector, which has motivated the industrial sector to innovate novel products, despite the challenges, including microbial stability, undesirable sensory impacts, genetic variability, and industrial scale-up. These challenges are now being addressed through advanced approaches, such as LAB genetic engineering techniques, including site-specific chromosomal deletions, mutations, stable integrations and insertions, introducing DNA into LAB cells, and the genome editing of LAB using the CRISPR–Cas system, as well as optimizing LAB fermentation through simultaneous saccharification and fermentation and enhancing LAB tolerance to acid through adaptive evolution. The future of LAB in beverage fermentation is bright, with advancements in technology, sustainability, and consumer-driven trends paving the way for innovative and health-enhancing products. Continued research and development will be crucial in overcoming the current challenges and maximizing the benefits of LAB fermentation.

## Figures and Tables

**Table 1 foods-14-02043-t001:** Summary of the role of lactic acid bacteria (LAB) in fermented dairy and non-dairy beverages.

Aspect	Dairy Beverages	Non-Dairy Beverages	Specific Traits	Bioactive Compounds/Metabolites Produced	Representative LAB Species	Type of Verification Method	Refs.
Preservation	LAB contribute to preservation by producing lactic acid, which inhibits spoilage microorganisms and pathogens.	LAB produce organic acids and antimicrobial compounds that inhibit spoilage microorganisms and pathogens.	Acid tolerance, antimicrobial activity	Lactic acid, acetic acid, bacteriocins, hydrogen peroxide	*Lactococcus lactis*, *Lactiplantibacillus plantarum*, *Pediococcus acidilactici*	In vitro, molecular tools	[42,43,44,45]
Flavor and aroma	LAB play a key role in the development of taste, texture, and aroma.	LAB enhance sensory quality by producing various volatile flavor and aroma compounds.	Esterase and proteolytic activity	Diacetyl, acetoin, ethyl acetate, aldehydes	*Leuconostoc mesenteroides*, *Lacticaseibacillus helveticus*, *L. plantarum*	In vitro, consumer sensory evaluation	[44,45,46,47,48,49]
Nutritional value	LAB enhance the nutritional profile by synthesizing essential metabolites and bioactive compounds.	LAB increase nutritional quality by producing bioactive compounds and enhancing the availability of nutrients.	Proteolysis, vitamin synthesis, phytate degradation	B-group vitamins (e.g., B2, B12), bioactive peptides	*Limosilactobacillus fermentum*, *Lacticaseibacillus casei*, *L. plantarum*	In vitro (antimicrobial activity assays), in vivo (nutritional bioavailability data)	[42,46,48,49,50]
Health benefits	LAB offer probiotic benefits, such as gut health, antimicrobial properties, and antioxidant potential.	LAB provide probiotic effects, such as antimicrobial activities, antioxidant properties, and cholesterol-lowering effects.	Probiotic potential, cholesterol assimilation, antioxidant enzyme production	Exopolysaccharides, conjugated linoleic acid, γ-aminobutyric acid	*Lacticaseibacillus rhamnosus*, *L. casei*, *L. plantarum*, *Lactobacillus acidophilus*	In vitro, in vivo (animal), RCT (Randomized controlled trials)	[50,51,52,53]
Fermentation	LAB are used as starter cultures to ensure efficient fermentation and product stability.	LAB are selected for their ability to ferment various plant matrices effectively, ensuring product stability and quality.	Rapid acidification, carbohydrate metabolism	Lactic acid, CO_2_, mannitol	*L. lactis*, *Streptococcus thermophilus*, *L. plantarum*, *Weissella cibaria*	In vitro (fermentation trials), consumer acceptability	[42,46,48,49,50]
Probiotic potential	Dairy LAB strains are often used for their probiotic properties, contributing to overall health benefits.	Non-dairy LAB strains are recognized for their probiotic potential, enhancing the health benefits of the beverages.	Bile salt tolerance, adhesion to intestinal cells	Exopolysaccharides, antimicrobial peptides	*Limosilactobacillus reuteri*, *Lacticaseibacillus paracasei*, *L. plantarum*, *Pediococcus pentosaceus*	In vitro (cell line studies), in vivo (animal), RCTs	[50,51,52,53]

**Table 2 foods-14-02043-t002:** Summary of some dairy, non-dairy, and hybrid beverages and associated lactic acid bacteria (LAB) functionalities.

Beverage Type	Key LAB Strains	Main Compound Production/Functionality	Refs.
Milk Kefir	*Lacticaseibacillus kefiranofaciens*, *Lentilactobacillus kefiri*, *Lacticaseibacillus kefirgranum*, *Lentilactobacillus parakefiri*, *Streptococcus thermophilus*, *Lactococcus* spp.	Production of kefiran (exopolysaccharide) with antimicrobial properties; synthesis of bioactive peptides and vitamins; flavor compounds, like acetaldehyde, ethanol, diacetyl, and acetoin	[54,57,58,59,60]
Buttermilk	*Lactobacillus* spp., *Lactococcus* spp., *Streptococcus* spp.	Enhancement of phospholipid content (e.g., L-α phosphatidylinositol, L-α phosphatidylcholine); increased levels of acetic and butyric acids; production of γ-aminobutyric acid (GABA); synthesis of folate and conjugated linolenic acid (CLA); generation of bioactive peptides with angiotensin-converting enzyme (ACE) inhibitory and mineral-binding activities	[67,69,70,71]
Yogurt drinks	*Lactobacillus delbrueckii* subsp. *bulgaricus*, *S. thermophilus*, *Lactobacillus acidophilus*, *Lactococcus lactis* subsp. *lactis*, *L. lactis* subsp. *cremoris*, *Lactiplantibacillus plantarum*, *Leuconostoc* spp.	Production of exopolysaccharides enhancing texture and stability; synthesis of bioactive compounds (e.g., vitamins, antioxidants); probiotic effects, including gut health improvement and immune modulation; flavor enhancement through incorporation of functional additives like tea and fig syrup	[77,78,79,80,81,83,85]
Vinegar	*L. plantarum*, *Limosilactobacillus fermentum*, *Lentilactobacillus buchneri*, *Lacticaseibacillus casei*, *Liquorilactobacillus acetotolerans*, *L. lactis*, *Pediococcus acidilactici*, *Pediococcus pentosaceus*, *Weissella confusa*, *Fructobacillus tropaeoli*, *Leuconostoc mesenteroides*, *Weissella paramesenteroides*, *Levilactobacillus* brevis, *Lacticaseibacillus* paracasei, *Lacticaseibacillus rhamnosus*, *Companilactobacillus paralimentarius*	Flavor enhancement; increased vitamin C, volatile organic compounds, and organic acids; possible roles in microbial stability and fermentation efficiency	[87,88,89]
Wine	*Oenococcus oeni*, *L. plantarum*, *Liquorilactobacillus mali*, *Liquorilactobacillus satsumensis*, *L. paracasei*, *Pediococcus parvulus*	Malolactic fermentation (malic acid to lactic acid), flavor development via volatile compounds, wine stabilization, haze and anthocyanin reduction, ochratoxin A removal, diacetyl production, spoilage (phenols, biogenic amines)	[2,40]
Beer	*L. plantarum*, *L. brevis*, *L. rhamnosus*, *L. acidophilus*, *Leuconostoc pseudomesenteroides*, *L. acetotolerans*, *Pediococcus damnosus*	Acidification (lactic acid), aroma development, probiotic viability, reduced pH, enhanced sensory properties, hop tolerance; spoilage risk (off-flavors, turbidity), control via yeast–LAB sequencing and brewing strategies	[115,116]
Milk–oat fermented beverage	*S. thermophilus*, *L. delbrueckii* subsp. *bulgaricus*, *L. plantarum* 299v, *L. acidophilus* La5	Enhanced sensory attributes depending on bacterial mix and base materials; combined nutritional benefits from dairy and oats	[7,143]
Milk–soy fermented beverage	*L. acidophilus* La5	Improved sensory properties; high consumer acceptability	[144]
Milk–quinoa fermented beverage	*L. acidophilus*	Enhanced phenolic content, antioxidant activity, mineral and amino acid levels; quinoa-stimulated microbial growth	[143]
Milk–watermelon seed yogurt	*Streptococcus salivarius* subsp. *thermophilus* EMCC104, *L. delbrueckii* subsp. *bulgaricus* EMCC1102	Enhanced antioxidant activity; improved renal function in hyperuricemic rats via increased enzymatic antioxidant defense; potential for upcycling food waste	[152]

**Table 3 foods-14-02043-t003:** Bioactive compounds produced by LAB, their food sources, chemical nature, and relevance to health.

Bioactive Compound	LAB Source	Chemical Nature	Beverage/Food Type	Health Relevance	Refs.
Peptides	Dairy products, various foods	Protein fragments	Fermented dairy products	Antimicrobial, antihypertensive, immunomodulatory	[155,156,157]
Exopolysaccharides (EPS)	Dairy products	Polysaccharides	Fermented dairy products	Antioxidant, cholesterol-lowering, prebiotic	[155,158]
Bacteriocins	Various foods	Antimicrobial peptides	Fermented foods	Antimicrobial	[155,159,160]
Lactic acid	Various foods	Organic acid	Fermented foods and beverages	Antimicrobial, preservative	[155,159,160]
Conjugated linoleic acid (CLA)	Dairy products	Fatty acid	Fermented dairy products	Cardiovascular health	[161]
Gamma amino butyric acid (GABA)	Plant-based beverages, grape juice	Non-protein amino acid	Fermented plant-based beverages	Antihypertensive, neuroprotective	[13,161,162]
Vitamins (e.g., folate, vitamin E)	Various foods	Vitamins	Fermented foods	Antioxidant, nutritional enhancement	[156,163]
Organic acids (e.g., acetic acid)	Rice bran, various foods	Organic acids	Fermented rice bran	Antioxidant, antimicrobial	[163]
Phenolic compounds (e.g., ferulic acid)	Rice bran, beetroot	Phenolic acids	Fermented rice bran, beetroot	Antioxidant	[163,164]
Diacetyl	Various foods	Organic compound	Fermented foods	Antimicrobial	[159,160]
Hydrogen peroxide	Various foods	Reactive oxygen species	Fermented foods	Antimicrobial	[159,160]
Reuterin	Various foods	Antimicrobial compound	Fermented foods	Antimicrobial	[159]

**Table 4 foods-14-02043-t004:** Comparison of the metabolic properties of homolactic and heterolactic metabolism of LAB.

Property	Homolactic Metabolism	Heterolactic Metabolism	Refs.
Primary pathway	Embden–Meyerhof pathway (EMP)	Phosphoketolase pathway (PKP)	[165,166,167]
Main products	Primarily lactic acid	Lactic acid, acetic acid, ethanol, and/or mannitol	[165,166,167]
Energy yield	Higher ATP yield per glucose molecule	Lower ATP yield per glucose molecule	[165,166]
Key metabolites	Acetoin, phenyllactic acid, ρ-hydroxyphenyllactic acid, glycerophosphocholine, succinic acid	Ornithine, tyramine, γ-aminobutyric acid	[165]
Fermentation characteristics	Rapid pH decline, higher lactate concentration, lower ethanol and ammonia production	Slower pH decline, higher acetate concentrations, and increased ethanol and ammonia production	[165,168]
Applications	Dairy fermentations (e.g., yogurt)	flavor compound production in dairy (e.g., cheese)	[166,169]
Strain examples	*Lactococcus lactis*, *Streptococcus thermophilus*	*Leuconostoc mesenteroides*, *Levilactobacillus brevis*	[165,166,169]
Sensory impact	Enhances flavor and aroma by producing lactic acid, which contributes to a tangy taste and can improve the overall sensory profile of beverages. Improves texture and mouthfeel by increasing viscosity and creaminess.	Produces a more complex flavor profile by generating a variety of metabolites, including ethanol, acetic acid, and carbon dioxide, which contribute to a diverse sensory experience. Adds carbonation to beverages, enhancing mouthfeel and providing a refreshing quality.	[170,171,172]
Functional/health impact	Promotes gut health by increasing the population of beneficial bacteria in the gut. Enhances the bioavailability of nutrients such as vitamins and amino acids. Boosts the immune system through the production of bioactive compounds. Decreases anti-nutritional factors, like phytic acid.	Increases antioxidant activity, which can help in reducing oxidative stress. May contribute to cardiovascular health by producing bioactive compounds that help in reducing cholesterol levels. Provides anti-inflammatory benefits through the production of various metabolites. Improves metabolic health by aiding in glycemic control and reducing the risk of diabetes	[170,171,172,173]

**Table 5 foods-14-02043-t005:** Advantages and limitations of some methods used for the identification of lactic acid bacteria (LAB).

Method	Advantages	Limitations	References
Phenotypic methods	Simple and cost-effective; useful for initial screening	May misidentify targets; time-consuming and labor-intensive	[201,204,205]
16S rDNA sequencing	High accuracy and reliability; can identify species and subspecies	Requires specialized equipment and expertise; can be expensive	[201,206]
PCR-based methods	High specificity and sensitivity; rapid and reliable	Potential for cross-reactions; requires specific primers and conditions	[179]
MALDI-TOF MS (Matrix-assisted laser desorption/ionization time-of-flight mass spectrometry)	High accuracy and efficiency; rapid identification	Limited by database quality; may not differentiate closely related species	[179,206]
TRFLP (Terminal restriction fragment length polymorphism)	Sensitive and can discriminate species in mixed cultures	Requires bioinformatics tools for analysis; may not be suitable for all LAB species	[207]
Near-infrared spectroscopy	Non-destructive and simple; high classification rates at genus level	Lower accuracy at species level; limited to specific spectral ranges	[202]
Polyphasic approach	Combines multiple methods for higher accuracy; reduces misidentification risk	More complex and time-consuming; requires integration of different data types	[200]

## Data Availability

The original contributions presented in this study are included in the article. Further inquiries can be directed to the corresponding author.

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
