# Peer review of "Highlighting Lactic Acid Bacteria in Beverages: Diversity, Fermentation, Challenges, and Future Perspectives"

_foods, 2025, doi:10.3390/foods14122043_

Round 1
Reviewer 1 Report
Comments and Suggestions for Authors
Manuscript 3626601
Journal Foods
Title Highlighting lactic acid bacteria in beverages: Diversity, fermentation, challenges, and future perspectives
The manuscript entitled “Highlighting lactic acid bacteria in beverages: Diversity, fermentation, challenges, and future perspectives” describes the role of LAB in influencing sensory characteristics, promoting health benefits, extending shelf life, and enhancing the safety of beverages. Emerging trends such as the use of LAB for the development of novel LAB-based beverages, their use for bioremediation of toxic compounds, genetic engineering of LAB strains to optimize and tailor their fermentation outcomes, and their use in drug delivery are also considered. The review is not original. Several reviews are available on this topic.
Several parts of the review should be improved/revised/completely rewritten. Please follow the comments in the file.

Author Response
Responses to reviewer 1 comments
Manuscript 3626601
Journal Foods
Title Highlighting lactic acid bacteria in beverages: Diversity, fermentation, challenges, and future perspectives
Comment 1
L32-39 Reduce this part to one third
Response 1
The part was reduced to one third as follows: “Lactic acid bacteria (LAB) are Gram-positive, non-spore-forming, generally non-motile bacteria classified under the order Lactobacillales (phylum Firmicutes). They exhibit cocci or rod shapes, thrive in both aerobic and anaerobic conditions, and play key roles in fermentation with applications in food, agriculture, and health sectors [1-4].”
Comment 2
L50-57 Which are the enzymatic activities involved in the fermentation of different beverages? Discuss this aspect in relation to the type of beverage (dairy or plant-based).
Response 2
This part was added to address the mentioned comment: “In the dairy fermentation, the enzymatic activities of LAB include glycolysis, proteolysis, lipolysis, and formation of various flavor compounds. Through glycolysis, LAB convert lactose and other sugars into lactic acid, reducing pH and acting as a preservative. For example, Streptococcus thermophilus and Lactobacillus delbrueckii ssp. bulgaricus are commonly used in dairy fermentations and have specific growth rates on different sugars [32]. The proteolytic enzymes of LAB break down milk proteins into peptides and amino acids, contributing to flavor and texture which is essential for the development of cheese and yogurt, where casein is hydrolyzed [33]. LAB lipolysis involves breakdown of milk fats into free fatty acids and other compounds, influencing flavor [33-34]. Moreover, LAB produce diacetyl, acetoin, and other volatile compounds through citrate metabolism, enhancing the flavor of products like cultured butter and sour cream [34-35].
The enzymatic activities of LAB in plant-based products include carbohydrate fermentation, phytase activity, proteolytic activity, and enzyme inhibition. LAB ferment simple sugars like glucose and sucrose, but not polysaccharides due to the lack of hydrolytic enzymes. Specific strains like Leuconostoc mesenteroides rapidly ferment sucrose, while others like Lactococcus lactis metabolize maltose and lactose [32,36]. Some LAB strains can improve mineral bioavailability in plant-based substrates due to their phytase activity that breaks down phytates which are anti-nutrient substances [37]. LAB can also hydrolyze plant proteins, producing bioactive peptides with health benefits. For example, Lacticaseibacillus rhamnosus showed high proteolytic activity in soy milk and produced soy bioactive peptides, which are beneficial to health [38]. LAB can synthesize compounds like γ-aminobutyric acid (GABA) from plant substrates, contributing to the health benefits of fermented beverages [13]. In addition, fermentation can increase the content of phenolic substances and antioxidants in plant-based beverages. LAB fermentation can result in the production of compounds that inhibit enzymes like α-glucosidase and α-amylase, which are beneficial for managing blood sugar levels [39].”
Comment 3
L82-86 Delete from the objective. It is not a primary goal of the review.
Response 3
It was deleted.
Comment 4
L97-98 Revise this sentence. It is not correct. Probably the family is divided into 25 genera. Please check.
Response 4
The information was checked and corrected as follows: “. The family Lactobacillaceae now includes 31 genera. The genus Lactobacillus was split into 25 genera including one retained genus (Lactobacillus) and 1 previously existing genus (Paralactobacillus), with 23 new genera introduced to accommodate species formerly classified under Lactobacillus.”
Please note that in 2020 the genus Lactobacillus was split into 23 new genera and in addition to Lactobacillus itself and a preexisting genus “Paralactobacillus” which make the total number 25 genera. The number of genera in Lactobacillaceae family was corrected to “31” (Ref. Zheng et al., 2020; A taxonomic note on the genus Lactobacillus: Description of 23 novel genera, emended description of the genus Lactobacillus beijerinck 1901, and union of Lactobacillaceae and Leuconostocaceae).
Comment 5
Table 1 Revise the table including specific traits, bioactive compounds, molecules and metabolites produced. Moreover, some species involved in these metabolic activities should be specified in the Table 1.
Response 5
Table 1 was revised by adding 3 columns for specific traits, bioactive compounds/metabolites produced, and representative species. In addition, the other reviewer recommended adding the “Type of verification method”, so this was added to Table 1 as a fourth column.
Comment 6
L106-127 Add the bioactive compounds released by LAB in kefir beverages and the development of innovative kefir beverages using non-conventional ingredients. See the literature to improve this part.
Response 6
This was added with new references: “LAB release several bioactive compounds in kefier beverages. Kefiran is a major exopolysaccharide produced by LAB such as L. kefiranofaciens in kefir, known for its antimicrobial properties. Exopolysaccharides were shown to inhibit pathogenic bacteria such as Listeria monocytogenes and Salmonella Enteritidis [57]. LAB produce bacteriocins and organic acids such as lactic acid and acetic acid that contribute to the antimicrobial properties of kefir. LAB also produce bioactive compounds due to their proteolytic activity on milk proteins (caseins and whey proteins). These peptides exhibit various health benefits, including antimicrobial, antihypertensive, and immunomodulatory effects [58-59]. LAB in kefir also synthesize certain vitamins and release amino acids during fermentation, enhancing the nutritional value of the beverage [59]. Moreover, they produce volatile organic compounds such as acetaldehyde, ethanol, diacetyl, and acetoin, contributing to the unique flavor and aroma of kefir [60].
Recently, innovative kefir beverages have been produced using non-conventional ingredients and new technologies. Incorporating fruit juices, plant extracts, and essential oils into kefir can enhance its antioxidant and functional properties. For instance, black carrot juice has been shown to produce water kefir-like beverages with high antioxidant activity and favorable sensory properties [61]. Likewise, a novel kefir beverage using date syrup, whey permeate, and whey has been developed, optimizing the formulation to achieve high antioxidant activities and acceptable organoleptic properties [62]. Moreover, some researchers formulated water kefir with Russian olive juice and optimized the process of production to maximize phenolic content, antioxidant activity, and microbial viability to enhance the probiotic properties [63]. To meet consumers’ preferences, coffee-flavored kefir has been developed and it showed promising results in terms of sensory acceptance and purchase intent, with high probiotic counts and antioxidant capacity [64]. Technological innovation have also been utilized in kefir production. For example, spray drying and encapsulation improved the stability and shelf life of kefir and maintained its functional properties. However, these techniques may impact the viability of beneficial microorganisms [65]. Moreover, kefir-containing snacks have been produced using 3D food printing, which could increase kefir consumption by offering attractive shapes and maintaining high probiotic content [66].”
Comment 7
L140-141 Add quantitative results and specific results
Response 7
Quantitative results and specific results were added as follows: “For example, the content of L-α phosphatidylinositol has doubled from 5.2.to 10.48 % of total phospholipids (PL) and the concentration of L-α phosphatidylcholine has significantly increased from 19.7 to 22.8 %PL. Likewise, the concentration of acetic acid has increased from 90.6 to 99.9 mg/100 g dry matter (DM) and the content of butyric acid reached 74.1 mg/100 g DM while it was not detected in control samples. The used bacteria included 31 LAB (genera: Lactobacillus, Lactococcus, and Streptococcus) and Bifidobacterium strains [67].”
Comment 8
L142 Bifidobacterium is not a LAB genera
Response 8
It was deleted and corrected throughout the paper to be presented as a separate genera from other LAB.
Comment 9
L144-163 Rewrite this part including the milk fermented with different LAB species and the effect of fermentation on bioactive compounds, health benefits, safety, shelf-life and so on. Acidophilus milk should be deleted because the review is focused on all LAB species and not only L. acidophilus.
Response 9
The part about acidophilus milk was deleted and this part was added to address the comment: “Thus, fermenting milk with different LAB species enhances its nutritional value, provides various health benefits, ensures safety through antimicrobial activity, and can extend the shelf-life with proper management of post-acidification. Among the bioactive active compound produced are γ-aminobutyric acid in which LAB species such as L. lactis, L. rhamnosus, and Lacticaseibacillus paracasei can significantly increase content in fermented milk [69]. LAB species like Lactobacillus delbrueckii, and S. thermophilus can produce folate and conjugated linolenic acid during fermentation and cold storage, enhancing the nutritional profile of the milk [70]. Probiotic LAB strains such as L. plantarum generate bioactive peptides with angiotensin-converting enzyme inhibitory and mineral-binding activities, contributing to cardiovascular health and mineral absorption [71]. Fermented dairy products exhibit hypocholesterolemic and antioxidant properties, which are beneficial for cardiovascular health [72]. The LAB strains in fermented milk can improve gut health by enhancing the intestinal microflora, reducing lactose intolerance, and preventing gastrointestinal infections [73]. Moreover, consumption of fermented dairy products can modulate the immune system and reduce allergic reactions [72]. LAB strains produce antimicrobial peptides and organic acids that inhibit pathogenic bacteria and fungi, enhancing the safety of fermented milk products [74]. Controlling post-acidification through strain selection and fermentation conditions is crucial because LAB continue to produce lactic acid during storage, which can affect the flavor and shelf-life of fermented milk [75]. Moreover, LAB strains such as L. plantarum and L. delbrueckii maintain high viability during storage, ensuring the continued probiotic benefits of the fermented milk [73].”
Comment 10
L168-170 Rewrite. It is not correct in English
Response 10
The sentence was rewritten to this: “Yogurt is produced by fermenting milk with L. delbrueckii subsp. bulgaricus and S. thermophilus. These lactic acid bacteria generate lactic acid, which causes milk proteins to coagulate, resulting in the characteristic texture and flavor of yogurt [77].”
Comment 11
L183-192 Enrich this part with other fermented yogurt drinks using LAB species
Response 11
This part was added with new references: “Fruit-flavored yogurts are made by adding fruit or fruit flavoring to enhance the sensory appeal and nutritional value of drinkable yogurts. Some researchers [81] formulated pomegranate and vanilla yogurt beverages that contained inulin as a prebiotic, along with probiotic bacteria Lactobacillus acidophilus and Bifidobacterium, to get symbiotic products. Another set of samples were supplemented with approximately 2 volumes of carbon dioxide. The formulated beverages were stabilized with high-methoxyl pectin and whey protein concentrate and compared to samples with added carbon dioxide. Both types of bacteria showed stability demonstrated by levels greater than 106 CFU/g for both flavors of beverage both with and without carbonation. These carbonated symbiotic drinkable yogurts have potential for commercialization. Moreover, drinkable yogurts can be made with functional additives by incorporating functional ingredients like coffee, tea, or plant extracts that can improve the bioactive properties and offer unique flavors to the products [82-84]. For instance, black and green tea were demonstrated to enhance the total phenolic content and antioxidant activity of yogurt significantly [83]. Likewise, addition of fig syrup was shown to enhance the sweetness and overall acceptability of drinkable yogurt [85]. These studies demonstrate a high potential to expand the variability of drinkable yogurts by incorporating various functional ingredients and commercializing them in the future.”
Comment 12
L204-206 Here and throughout the manuscript use the updated taxonomy of LAB species…for example Lactobacillus fermentum should be replaced by Limosilactobacillus fermentum. Revise
Response 12
The LAB species names were revised throughout the manuscript.
Comment 13
L213-252 Reduce this part to one third. This is a review, not an original research article. I strongly suggest to delete Figure 1 and 2. It is important for the acceptance of the manuscript
Response 13
Figures 1 and 2 were deleted, and the content (L213-252 ) was reduced to one third.
Comment 14
L334-341 Enrich the part focused on the role of LAB in the production of innovative beers. Section 2.2.2 is more focused on wine.
Response 14
An additional part was added (also in response to the other reviewer’s comments), including some novel beer production and challenges, including new references as follows: “Recent advancements in alcoholic beverage fermentation have explored the utilization of non-traditional cultures, including non-Saccharomyces yeasts and LAB, to contribute to unique flavors and improve the sensory characteristics of the final product [108]. Specific non-Saccharomyces yeasts like Torulaspora delbrueckii and Candida species have shown potential in enhancing fermentation processes and product quality [109]. The incorporation of probiotic strains and health-promoting compounds in alcoholic beverages is gaining traction. This approach aims to offer additional health benefits, such as improved gut health, alongside traditional alcoholic consumption. African traditional alcoholic beverages, for instance, are noted for their probiotic properties due to the active presence of LAB and yeasts during fermentation [110]. Moreover, advances in molecular biology and sequencing technologies are facilitating the identification and utilization of novel microbial strains. Techniques like whole-genome sequencing and third-generation DNA sequencing are being employed to better understand and harness microbial diversity [111].
Immobilized kefir culture is an innovative technique being explored in winemaking, particularly for producing low alcohol wines. This method involves the physical confinement of kefir cells on various supports, which can enhance fermentation efficiency and product quality [112]. For instance, low alcohol wines (≤10.5% vol) were produced using wet and freeze-dried immobilized kefir culture on natural supports. In comparison with the conventional free cells culture, the immobilized kefir culture showed high operational stability and it was found suitable for simultaneous alcoholic and malolactic low alcohol wine fermentation at temperatures up to 45°C with high ethanol productivity (up to 55.3 g/(Ld) and malic acid conversion rates (up to 71.6%). The produced wine was considered of high quality by the sensory panel, suggesting a potential industrial use in semi-dry low alcohol wine-making at 37°C and in producing novel wine products with a sweet (liquoreux) property at 45°C, which is advantageous in regions with tropical climates or hot summer periods [113].”
“LAB can also be used to create probiotic beers, which offer health benefits in addition to unique flavors. These beers are formulated with probiotic LAB strains that survive the brewing process and remain viable in the final product. For instance, researchers produced probiotic-enriched Gueuze-style sour beer utilizing a two-step fermentation process that involved alcoholic fermentation utilizing Saccharomyces boulardii CNCM 1-745 and lactic acid fermentation utilizing L. acidophilus LA3, L. acidophilus LA5 , L. plantarum 299v, L. rhamnosus GG, and L. pseudomesenteroides BIOTEC011. LAB fermentation enhanced the content of organic acids and volatile compounds, and the sensory characteristics. Moreover, high levels of probiotics were retained under different conditions, such as carbonation, storage at 4°C, and simulated gastrointestinal digestion [118]. A novel type of beer was produced using olive leaves as an ingredient. Olive leaves significantly increased the polyphenol content of beers, while their addition did not influence the antioxidant activity. About 5 g/L of olive leaves resulted in a beer with a pleasant sensory profile [119]. This kind of research opens the door for utilizing new ingredients for beer manufacturing, particularly adding a nutraceutical value.
However, while LAB are beneficial in controlled fermentations, they are also the most common spoilage bacteria in beer, capable of producing off-flavors (lactic acid, diacetyl) and turbidity, rendering the beer undrinkable. L. brevis was identified as a common spoiler in bottled microbrewed beer from Australia [120]. Other LAB, such as Liquorilactobacillus acetotolerans, L. plantarum, and Pediococcus damnosus were described as beer-spoiling bacteria [121]. Therefore, effective management of LAB strains and brewing conditions is essential to prevent spoilage. LAB used in beer production must be hop-tolerant, as hop acids can inhibit many bacterial strains. Specific LAB strains, such as L. plantarum and L. brevis, have been identified for their hop tolerance and suitability for beer fermentation [117,122].”
Comment 15
L359, L373, L379, L389 and so on. Bifidobacterium is not a LAB genera. Revise
Response 15
Bifidobacterium mention was checked throughout the manuscript and deleted when it was mentioned as LAB. It is now mentioned as a non-LAB genus.
Comment 16
L397-398 Add a part on the production of fermented legume beverages by using LAB species. The papers doi.org/10.3390/foods11223578 and doi.org/10.3390/foods11213346 are suggested for your analysis and discussion.
Response 16
The part of the production of fermented legume beverages by using LAB species was added and also the recommended references as follows: “The production of fermented legume beverages using LAB is a growing area of interest due to the health benefits and enhanced nutritional properties these beverages offer. The fermentation process typically involves mixing legumes with water and other ingredients, inoculating with LAB, and fermenting under controlled conditions (e.g. specific temperatures and pH levels). For instance, a study on wheat germ and sweet-waxy maize used a combination of L. bulgaricus, S. thermophilus, and Bifidobacterium lactis, fermenting at 38°C and pH 7.0 for 24 hours. The produced beverage tasted slightly sweet and sour and had refreshing flavor of wheat germ and sweet-waxy maize [129]. Additional processing steps, such as heat treatment and the addition of stabilizers, may be required to ensure the long-term storage and consistent quality of the fermented legume-based beverages [49,129]. LAB fermentation improves the nutritional profile of legume-based beverages by increasing the bioavailability of essential nutrients such as amino acids, minerals, and vitamins. It also leads to the accumulation of bioactive compounds like exopolysaccharides, short-chain fatty acids, and bioactive peptides, which contribute to the health benefits of these beverages [130]. Moreover, LAB fermentation effectively reduces antinutritional factors in legumes, such as trypsin inhibitors, cyanide, saponins, raffinose series oligosaccharides, tannins, and phytates. This reduction enhances the digestibility and nutritional value of the legumes [131]. Fermented legume beverages can serve as synbiotic foods, combining probiotics (beneficial bacteria) and prebiotics (compounds that support the growth of beneficial bacteria). This is because legumes naturally contain prebiotic ingredients like oligosaccharides, resistant starch, polyphenols, and isoflavones, which support the growth of LAB [130].
Various LAB strains are used in the fermentation of legume beverages, including L. plantarum, L. lactis, and other Lactobacillus species. These strains are selected for their ability to produce organic acids, antimicrobial substances, and other beneficial metabolites [132]. A study found that L. pseudomesenteroides significantly catabolize raffinose, maltose, and citrate, which are present in soy beverages, while L. lactis produced high concentrations of diacetyl and lactic acid, which are relevant for the generation of plant dairy alternatives. It also decomposed phytic acid, pectin, and sucrose, mostly present in bean, cereal, and fruit-based plant matrices [133]. Likewise, fermentation of an Apulian black chickpeas protein concentrate using S. thermophilus alone, and its co-cultures L. lactis and L. plantarum yielded beverages with high protein (120.00 g/kg) and low-fat (17.12 g/kg) contents, while the levels of phytic acid decreased and saturated fatty acids largely decreased. The formulated beverages had greater lightness, consistency, cohesivity, and viscosity than the controls. Interestingly, the aromas of legumes and grass were not evident in these beverages, possibly due to the formation of new volatile organic molecules [134]. Another study fermented the water extracts of lupin, pea, and bean grains by inoculating them with L. acidophilus ATCC 4356, L. fermentum DSM 20052, and L. paracasei subsp. paracasei DSM 20312. Fermentation of bean water extract resulted in an unpleasant ferric-sulfurous off-odor. However, lupin and pea legume-based beverages had improved sensory characteristics and retained high levels of viable LAB until the end of the cold storage [135].”
Comment 17
L397-398 Add a part on the production and functional characteristics of cereal-based beverages started with lactic acid bacteria.
Response 17
The part was added with new references as follows: “Cereal-based beverages fermented with LAB have gained popularity due to their functional and nutritional benefits. The production process typically involves the fermentation of cereal substrates by LAB, which can be done using monocultures or co-cultures with other microorganisms such as yeasts. Common cereals used include malt, rice, maize, barley, and buckwheat. LAB such as L. rhamnosus, L. acidophilus, L. plantarum, and Lactobacillus helveticus are frequently used. These bacteria can be used alone or in combination with yeasts like S. cerevisiae [136-140]. Enzymes like α-amylase, protease, and glucoamylase are often used to hydrolyze the cereal substrates, making sugars and amino nitrogen more available for bacterial growth [137,138]. Moreover, parameters such as temperature, pH, and fermentation time are optimized to enhance bacterial growth and product quality [138]. The fermentation process enhances the functional potential of cereal-based beverages. Fermented cereal beverages are rich in organic acids, free amino acids, and bioactive compounds like gamma-aminobutyric acid [136-138]. These beverages often contain high counts of viable LAB, which can confer probiotic benefits, including improved gut health and enhanced immune function [139]. In addition, the fermentation process improves the flavor, aroma, and texture of the beverages. LAB contribute to the development of desirable sensory attributes by producing volatile compounds and organic acids [137,138,140-142]. Similar to other LAB fermented beverages, the acidic environment created by LAB fermentation inhibits the growth of pathogenic bacteria, thereby enhancing the safety and shelf life of the beverages [142]. Multi-Cereal Beverages can also be produced by combining different cereals like malt, rice, and maize, fermented with LAB and yeasts, yielding beverages with unique flavors and high nutritional value [138]. Thus, cereal-based beverages fermented with LAB offer a promising avenue for developing functional foods with enhanced nutritional and sensory properties. The use of specific LAB strains and optimization of fermentation conditions are crucial for producing high-quality beverages that meet consumer demands for health and taste [13,136-138].”
Comment 18
L399-414 Add other references related to hybrid beverages. The bioactive compounds and the health benefits of these beverages should be described
Response 18
More references were added along with some bioactive compounds and the health benefits. However, this part is less discussed in the literature than diary and non-dairy products alone. This was added as follows: “Hybrid yoghurt was produced by blending cow's milk with soy and oat drinks in various ratios. The hybrid yogurt resulted in improved viscosity, favorable pH gradient, and the absence of pathogens in the final product, demonstrating the microbial safety of the products. This approach in hybrid yogurt production can enhance consumer acceptance by combining dairy and plant-based derivatives [150]. Hybrid fermented beverages with LAB are rich in various bioactive compounds produced during fermentation. For instance, a fermented milk product was prepared by mixing cow’s milk and quinoa beverage with a starter culture containing L. acidophilus and Bifidobacterium bifidum). The addition of quinoa beverage stimulated the growth of the starter culture and yielded a final product with higher total phenolic content, minerals, antioxidant activity, and amino acids than control [143]. An attempt was also made to produce kefir via the addition of quinoa flour or rice flour to cow’s milk. The fortification of kefir with quinoa flour reduced the fermentation time 2.5 hours, while kefir fortified with rice flour reduced it 1.5 h in comparison with the control. However, the controls received better acceptance by the panelist, probably due to their better acidity. The kefir fortified with 0.5% quinoa obtained the highest viscosity and acidity. The reduced fermentation time was due to the addition of prebiotics from quinoa and rice flours that are sources for proteins and polysaccharides [151].
A new yoghurt was prepared from cow’s milk blended with watermelon seed milk and inoculated with Streptococcus salivarius subsp. thermophilus EMCC104 and L. delbruekii subsp. bulgaricus EMCC1102. Hyperuricemic rats fed a diet supplemented with 10% watermelon seed milk yoghurt showed a significant improvement in renal function compared to the control group. This effect could be due to the increased antioxidant activity via enhancement of the functions of superoxide dismutase, catalase, and glutathione transferase. This example shows that byproducts of food waste can be combined with dairy materials to produce novel hybrid beverages with enhanced functional and nutritional properties [152]. Another study produced a functional yogurt drink fortified with golden berry juice and examined its therapeutic effect on hepatitis rats. The yogurt drinks fortified with golden berry juice had the highest content of total phenolic compounds, antioxidant activity, and organoleptic scores compared to the controls. Rats fed on a diet fortified with functional yogurt drinks containing golden berry juice for 8 weeks showed higher potential hepatoprotective effects compared with the liver injury control group, highlighting the potential of using this hybrid drink in protecting the liver [153].”
Comment 19
L416-536 Rewrite this section. The description of the metabolism of LAB is out of the scope of this review. Author should describe the metabolism of LAB during beverage fermentation. Revise this part describing the metabolic activities of LAB during beverage fermentation and add specific examples. Moreover, this part could be reduced in length
Response 19
This section was rewritten. Its title has been changed to “ Metabolic Activities of LAB During Beverage Fermentation”. It was also shortened. The discussion now has been linked to beverages and the previous part was abbreviated but also utilized and combined with new references and information and new sections were produced instead of the previous ones. The previous subsections homo and heterofermentation were combined under carbohydrate metabolism along with sugar fermentation. Another subsection of “organic acid metabolism” was added and the part of malolactic fermentation was combined with it and shortened. The part of “Respiration metabolism” was deleted to shorten this part. Some information from “Amino acid catabolism” was combined with the newly added subsection “Amino Acid and Protein Metabolism”, the section was shortened with the information related to flavor development was retained. The part about “Polyunsaturated fatty acid (PUFA) metabolism” was shortened and new references that involve beverages were added. Please refer to the manuscript for the detailed description.
Comment 20
L538-579 Rewrite this section. The isolation and identification of LAB (starters and autochthonous LAB) from fermented beverages should be described. The NGS techniques should be also included
Response 20
The section was rewritten and restructured into this format: Sample collection, LAB isolation, LAB identification utilizing conventional phenotypic, biochemical and other methods, then molecular and NGS giving examples of their utilization in identifying LAB in beverages. Other modifications in this part were done according to the other reviewer’s comments.
Comment 21
L630-652 Delete. This part is not in line with the objective of the review.
Response 21
This part was deleted.
Comment 22
L580-687 Include in this section the fermentation of beverages derived from food by-products or food waste extracts as ingredients. The adaptation of LAB species to these fermentations can promote the circular economy
Response 22
The information was included as follows: “Moreover, the production of LAB fermented beverages can provide means for promoting the circular economy through utilization of agro-industrial wastes and non-conventional sources such as fruit pulps and peels, potato peels, and faba bean [222]. Byproducts from the beverage industry, such as spent grains from brewing and pomace from winemaking, are also utilized. These contain valuable bioactive compounds like polyphenols and dietary fibers that can enhance the functionality of the fermented beverages [223]. For instance, watermelon seed milk was utilized in yoghurt manufacturing and was demonstrated to improve kidney function as an anti-hyperuricemic agent [152]. LAB have various adaptation mechanisms that permit them to utilize various carbon sources, including sugars and organic acids, which are abundant in food by-products and waste extracts. For instance, LAB convert L-malic acid to L-lactic acid during malolactic fermentation in wines and ciders, enhancing flavor and stability. They can also metabolize glycosides and other carbon sources, contributing to the sensory quality of the fermented products. LAB can thrive in various environmental conditions, including different pH levels and temperatures. For instance, L. plantarum can survive at high cell densities in orange juice at 4°C without altering its organoleptic properties, making it suitable for functional juice production [224]. On the genetic level, LAB have undergone gene loss and gain events, allowing them to adapt to nutrient-rich niches and dominate specific habitats. This genomic plasticity is crucial for their survival and efficiency in diverse fermentation environments [225,226].”
Comment 23
L707-710 Delete this part. It is not important in the conclusion section
Response 23
This part was deleted
Comment 24
L726-1142 Revise the references according to previous comments. Revise the numbering in the main text
Response 24
The references were revised. Now there are 248 references.

Reviewer 2 Report
Comments and Suggestions for Authors
In this work, the author discusses lactic acid bacteria in beverages and examines their diversity, fermentation, challenges, and future perspectives. The topic is interesting and the manuscript could have a potential from a scientific point of view. However, there are major issues that require the author’s attention.
Introduction:
- Line 43: I believe a paragraph presenting information about market trends, sales and other important data regarding “Lab containing foods” can be created in this section. As an alternative, this information can be presented in a small table for all product categories discussed in this review.
- Lines 42-47: The health-promoting character of functional drinks has to be better presented.
- Maybe a small paragraph about legislation and regulations (e.g. about GRAS strains vs. novel genetically modified LAB) would also be handy if presented in the introduction section or even later on in the discussion.
- The aim paragraph seems intriguing. I am not sure, however, if this hybrid system is ideal for this kind of work. I would prefer the authors to follow a more classical approach for the manuscript, meaning focus to a narrative review or focus to the presentation and discussion of the 5 LAB species identified in vinegar. Trying to cover both might lead to under-representation of the hands on work (for the vinegar). In any case, the aim paragraph has to be edited, for clarity reasons, in order to present in a better way these 2 objectives of the manuscript.
Table 1 is quite informative. I believe it should be expanded and incorporate one more column with information regarding the experimental status of each aspect. For instance was the aspect verified with experiments in vitro or in vivo, RCTs, consumers, etc. In this way, the robustness of each aspect will be better presented.
The information presented in the following sections are interesting and lay the path for a broader understanding regarding the fermented foods. However, at this form, it is difficult for the reader to find the key information in each subsection. Maybe the author should create a small table in each subsection (or a bigger one), where key strains for each beverage type (e.g., kefir, kombucha, vinegar) will be linked to their main compound production (e.g. lactic acid production, diacetyl production, ethanol production etc) or to their functionality (ideally probiotic).
Lines 288-291: The authors should also refer to emerging research trends regarding the use of non-traditional cultures or microbial groups in alcoholic beverages production (e.g. https://doi.org/10.1002/jsfa.10363, https://doi.org/10.3390/fermentation7020045, etc).
In Chapter 3:
- Homo‑ and hetero-fermentation would be easier to present in a small figure. It is better for visualization.
- Table 3 can be edited and expanded (or a new one created). In my opinion, it would be useful to link each pathway to its sensory or functional outcome. This is important for pinpointing the effect on the consumers’ perception and/or health impact (if any).
In Chapter 4: Again, better to create a small table and present pros/cons of each technique and approach. These aspects, however, have already been presented in multiple other manuscripts. I believe the author should focus on emerging techniques (e.g. TOF MS, etc) and also discuss limitations of other long-used techniques, in this section.
Conclusions should be improved after addressing my comments above. In general, start with a summary of results, and finalize with an outlook. Future plans and directions can also be showed here. Again I have to say that I am deeply troubled by this hybrid approach of this manuscript. Personally, I would present the lab vinegar results in a separate manuscript.
Comments on the Quality of English Language
Minor editing.
Author Response
Reviewer 2
Introduction:
Comment 1
- Line 43: I believe a paragraph presenting information about market trends, sales and other important data regarding “Lab containing foods” can be created in this section. As an alternative, this information can be presented in a small table for all product categories discussed in this review.
Response 1
This paragraph has been added: “This expansion has been driven by the growing consumer awareness and positive perception towards probiotics in beverages and other food products, as seen in Malaysia where over 80% of consumers are knowledgeable about the benefits of probiotics [9]. Currently, LAB-containing foods are widely available in global markets, including China, Germany, Jordan, Korea, Lithuania, New Zealand, Poland, Singapore, Thailand, Turkey, and Vietnam. Probiotic strains have been used in these markets for decades without adverse events [10]. The global LAB market was valued at USD 1.16 billion in 2024, and is projected to reach USD 2.22 billion by 2034, growing at a CAGR (Compound Annual Growth Rate) of 6.1% [11]. The Asia-Pacific region is leading the global market and is expected to grow at a CAGR of 9.2%, driven by a strong cultural desire towards fermented food and beverages and increasing health consciousness in countries like China, Japan, and South Korea. The regions of North America and Europe are showing steady growth, with North America projected at a CAGR of 7.5% and Europe at 7.8%, supported by high consumer awareness and a strong focus on health and wellness products [12].”
Comment 2
- Lines 42-47: The health-promoting character of functional drinks has to be better presented.
Response 2
The paragraph about health benefits was expanded with the addition of new references as follows: “Our knowledge about the health benefits of LAB is expanding due to the probiotic properties of many strains. Examples of these benefits include reduction of lactose intolerance and cholesterol levels, vitamin synthesis, and enhancement of gut health [4,21,23]. LAB help maintain the balance of intestinal microflora by suppressing harmful bacteria, which is crucial for gut health. LAB fermentation enhances antioxidant activity in functional drinks. This is achieved through the production of bioactive compounds such as phenolics and flavonoids [24]. They also have immunomodulatory and antitumor properties [25]. LAB produce bacteriocins and other antimicrobial compounds that inhibit the growth of pathogenic bacteria, contributing to the safety and shelf-life of the functional drinks [14]. Moreover, LAB may help reduce cardiovascular diseases by producing bioactive compounds that have beneficial effects on heart health [26]. LAB-fermented drinks can play a role in managing metabolic diseases through their anti-inflammatory effects and ability to balance microbiota [27].The level of LAB should be 106 CFU/mL/g or more to exhibit the probiotic or nutraceutical functional properties [28,29]”.
Comment 3
- Maybe a small paragraph about legislation and regulations (e.g. about GRAS strains vs. novel genetically modified LAB) would also be handy if presented in the introduction section or even later on in the discussion.
Response 3
This paragraph was added to the discussion because related information is there as follows: “This is because even though most LAB strains are “generally recognized as safe” (GRAS) by the US FDA (Food and Drug Administration) and as having Qualified Presumption of Safety (QPS) by the European Food Safety Authority [228], using them in new products can lead to the production of substances for which safety should be checked [229,230]. Genetically modified LAB (GM-LAB) face stricter regulatory scrutiny compared to traditional LAB strains. This is primarily due to concerns about the dissemination of antibiotic resistance genes and the environmental impact of GM organisms [230]. One of the major concerns with GM-LAB is the potential for horizontal gene transfer of antibiotic resistance markers, which can pose significant public health risks. Regulatory bodies require thorough evaluation of antibiotic resistance profiles and the potential for gene transfer before approving GM-LAB for use [231]. To mitigate risks, strategies such as biocontainment, use of food-grade selection markers, and homologous DNA are recommended. These measures aim to prevent the release of GM-LAB into the environment and ensure their safe use in closed systems [230].”
Comment 4
- The aim paragraph seems intriguing. I am not sure, however, if this hybrid system is ideal for this kind of work. I would prefer the authors to follow a more classical approach for the manuscript, meaning focus to a narrative review or focus to the presentation and discussion of the 5 LAB species identified in vinegar. Trying to cover both might lead to under-representation of the hands on work (for the vinegar). In any case, the aim paragraph has to be edited, for clarity reasons, in order to present in a better way these 2 objectives of the manuscript.
Response 4
According to reviewer 1: The original information about LAB in date vinegar was deleted including Figure 1 and 2 and the information in the supplementary file (they will be utilized in a separate manuscript). Therefore, only the types of LAB in vinegar were retained. This information was also deleted from the abstract, objectives and the conclusion. The review follows now the classical model of not including large new results.
“Given the rapid growth of the functional beverage sector, this review aims to summarize the current state of knowledge regarding LAB in beverages, emphasizing their diversity across dairy, non-dairy, and hybrid beverage matrices. This comparative approach helps readers understand how LAB behave and contribute across different substrates. Details about LAB fermentation processes, associated challenges, and prospects are also discussed to assist bridging knowledge gaps and identifying research opportunities in the rapidly growing functional beverage sector. Further, it discusses their possible contribution to the fermentation process, emphasizing the necessity to explore the LAB present in unexplored food matrices and utilize them to manufacture novel beverages and foods, possibly valorizing low-value raw materials.”
Comment 5
Table 1 is quite informative. I believe it should be expanded and incorporate one more column with information regarding the experimental status of each aspect. For instance was the aspect verified with experiments in vitro or in vivo, RCTs, consumers, etc. In this way, the robustness of each aspect will be better presented.
Response 5
A column for “Type of verification” was added to the table. In addition, 3 more columns were added to Table 1 in response to Reviewer 1 suggestions (These include Specific traits, Bioactive compounds/metabolites produced, and Representative LAB species).
Comment 6
The information presented in the following sections are interesting and lay the path for a broader understanding regarding the fermented foods. However, at this form, it is difficult for the reader to find the key information in each subsection. Maybe the author should create a small table in each subsection (or a bigger one), where key strains for each beverage type (e.g., kefir, kombucha, vinegar) will be linked to their main compound production (e.g. lactic acid production, diacetyl production, ethanol production etc) or to their functionality (ideally probiotic).
Response 6
A new table (Table 2) was generated to summarize some examples of LAB beverages and functionality of LAB in these beverages. This will allow the reader to get a summary of key beverage types mentioned in the paper.
Comment 7
Lines 288-291: The authors should also refer to emerging research trends regarding the use of non-traditional cultures or microbial groups in alcoholic beverages production (e.g. https://doi.org/10.1002/jsfa.10363, https://doi.org/10.3390/fermentation7020045, etc).
Response 7
These paragraphs were added: “Recent advancements in alcoholic beverage fermentation have explored the utilization of non-traditional cultures, including non-Saccharomyces yeasts and LAB, to contribute to unique flavors and improve the sensory characteristics of the final product [108]. Specific non-Saccharomyces yeasts like Torulaspora delbrueckii and Candida species have shown potential in enhancing fermentation processes and product quality [109]. The incorporation of probiotic strains and health-promoting compounds in alcoholic beverages is gaining traction. This approach aims to offer additional health benefits, such as improved gut health, alongside traditional alcoholic consumption. African traditional alcoholic beverages, for instance, are noted for their probiotic properties due to the active presence of LAB and yeasts during fermentation [110]. Moreover, advances in molecular biology and sequencing technologies are facilitating the identification and utilization of novel microbial strains. Techniques like whole-genome sequencing and third-generation DNA sequencing are being employed to better understand and harness microbial diversity [111].
Immobilized kefir culture is an innovative technique being explored in winemaking, particularly for producing low alcohol wines. This method involves the physical confinement of kefir cells on various supports, which can enhance fermentation efficiency and product quality [112]. For instance, low alcohol wines (≤10.5% vol) were produced using wet and freeze-dried immobilized kefir culture on natural supports. In comparison with the conventional free cells culture, the immobilized kefir culture showed high operational stability and it was found suitable for simultaneous alcoholic and malolactic low alcohol wine fermentation at temperatures up to 45°C with high ethanol productivity (up to 55.3 g/(Ld) and malic acid conversion rates (up to 71.6%). The produced wine was considered of high quality by the sensory panel, suggesting a potential industrial use in semi-dry low alcohol wine-making at 37°C and in producing novel wine products with a sweet (liquoreux) property at 45°C, which is advantageous in regions with tropical climates or hot summer periods [113].”
In Chapter 3:
Comment 8
- Homo‑ and hetero-fermentation would be easier to present in a small figure. It is better for visualization.
Response 8
Reviewer 1 highlighted a concern to reduce this part and to discuss the concept of homo‑ and heterofermentation in beverages, therefore, this section was rewritten with new subheadings and the fermentation types were discussed in relation with fermentation of LAB in beverages. The table was expanded to include sensory and functional outcomes as recommended in the next comment. The complicated information about the pathways were deleted and no new figure was drawn because Reviewer 1 advised to shorten this section and emphasized that a deep description of the metabolic pathways of LAB is beyond the objectives of this review. I think the new additions (2 rows) to Table 4 (sensory impact and functional/health impact) clarify very well the homolactic and heterolactic metabolism of LAB in relation to beverages, which is important for this review to highlight. However, if this figure is still necessary for this review, I can add it.
Comment 9
- Table 3 can be edited and expanded (or a new one created). In my opinion, it would be useful to link each pathway to its sensory or functional outcome. This is important for pinpointing the effect on the consumers’ perception and/or health impact (if any).
Response 9
The comment was addressed by adding two new rows (sensory impact, functional/health impact) to table 3 (Table 4 now). This allowed linking each pathway to its sensory or functional outcome.
Comment 10
In Chapter 4: Again, better to create a small table and present pros/cons of each technique and approach. These aspects, however, have already been presented in multiple other manuscripts. I believe the author should focus on emerging techniques (e.g. TOF MS, etc) and also discuss limitations of other long-used techniques, in this section.
Response 10
A table (Table 5) was added with new references. Title: “Advantages and limitations of some methods used for identification of lactic acid bacteria (LAB)”.
A new paragraph about MALDI-TOF-MS was added with example from beverages and new references as follows: “The automated machines, such as VITEK 2 compact, can be utilized for the identification of some LAB, while other more advanced machines, such as MALDI-TOF MS (Matrix-assisted laser desorption/ionization time-of-flight mass spectrometry) can provide identification of more LAB species, depending on the coverage of their databases. MALDI-TOF MS can quickly identify bacteria by analyzing their protein fingerprints, making it suitable for high-throughput applications. The method involves constructing extensive identification databases and using peak-based numerical analysis to achieve species-level identification. MALDI-TOF MS has been successfully used to identify beer spoilage bacteria such as L. brevis and P. damnosus. This is crucial for quality control in breweries to prevent microbial outbreaks that can lead to product recalls. Challenges in using MALDI-TOF MS include identifying bacteria in mixed cultures due to compatibility issues. However, novel approaches combining inertial microfluidics and secondary flows have been proposed to separate and identify spoilage microorganisms from mixed cultures efficiently [208,209]. In addition, the accuracy of MALDI-TOF MS depends on the comprehensiveness of the reference database. Additional reference strains may be necessary to increase sensitivity and specificity for species-level identification [210].”
Another new paragraph about new techniques was added as follows: “High-throughput sequencing (HTS) of the 16S rRNA gene can also be used to analyze the diversity and dynamics of LAB during fermentation processes. Some researchers analyzed the diversity and dynamics of LAB in atole agrio; a traditional maize-based beverage of Mexican origin. The HTS of the 16S rRNA gene confirmed the predominance of Lactobacillaceae and Leuconostocaceae at the beginning of the fermentation process. This technique also revealed that L. plantarum predominated in the liquid batches, whereas W. confusa was the dominant species in the solid batches [217]. Many LAB are identified and examined to test their potential use as probiotics or starter cultures. In one study [218], six LAB strains (L. fermentum 73B, P. pentosaceus 74D, L. fermentum 44B, W. confusa 44D, L. fermentum 82C, and Weissella cibaria 83E) that were isolated from spontaneously fermented Ethiopian cereal-based beverages (naaqe and cheka) were demonstrated to have antipathogenic, immunostimulatory, and starter culture potentials. Authors concluded that these bacteria can be used as autochthonous probiotic starters for naaqe and cheka fermentations after confirming their safety. Other high-throughput sequencing technologies, such as PacBio SMRT and Illumina MiSeq, are used to analyze the entire microbial community in fermented beverages. These methods can identify a wide range of LAB species and their relative abundances [219,220].”
This paragraph with new references was added to discuss the limitations of the conventional methods used for LAB identification: “The conventional methods have various limitations. Phenotypic methods can sometimes misidentify LAB due to the overlapping characteristics among different species [201]. These methods often require extensive sample preparation and can be time-consuming, making them less efficient for rapid identification needs. Conventional phenotypic methods often involve destructive testing, which can limit the ability to perform further analyses on the same sample [202]. Moreover, the phenotypic methods may not be sensitive enough to detect LAB in mixed cultures or under stressful conditions where bacteria may enter a viable but non-culturable (VBNC) state [203].”
Comment 11
Conclusions should be improved after addressing my comments above. In general, start with a summary of results, and finalize with an outlook. Future plans and directions can also be showed here. Again I have to say that I am deeply troubled by this hybrid approach of this manuscript. Personally, I would present the lab vinegar results in a separate manuscript.
Response 11
The original part about the results of date vinegar was deleted from the whole review paper. The conclusion was rewritten to summarize the results, then addressing the current approaches for challenges and finally, an outlook sentence as follows: “LAB have different metabolic pathways such as carbohydrate metabolism, organic acid metabolism, protein metabolism, and polyunsaturated fatty acid (PUFA) metabolism. LAB fermentation enhances the flavor profile of beverages by producing organic acids such as lactic acid that contribute to a tangy taste. The type of LAB and substrate used can significantly impact the texture and stability of fermented beverages. LAB can metabolize phenolic acids, resulting in bioactive metabolites, which suppress pro-inflammatory cytokines. LAB fermentation increases the antioxidant activity of beverages, enhancing their health-promoting properties. Bacteriocins and organic acids produced by LAB can inhibit pathogenic microorganisms, contributing to food safety and health benefits. LAB fermentation can enrich beverages with bioactive compounds like γ-amino butyric acid (GABA), which has various health benefits such as antihypertensive effects, neuroprotective and cognitive benefits, metabolic and endocrine benefits, and mental health benefits. The proven health benefits of fermented beverages, including dairy, non-dairy, and hybrid beverages, have stimulated and attracted more consumers toward this sector, which has motivated the industrial sector to innovate novel products, despite the challenges, including microbial stability, undesirable sensory impacts, genetic variability, and industrial scale-up. These challenges are now being addressed through advanced approaches such as LAB genetic engineering techniques including site-specific chromosomal deletions, mutations, stable integrations, and insertions and introducing DNA into LAB cells, and genome editing of LAB using CRISPR–Cas system as well as optimizing LAB fermentation through simultaneous saccharification and fermentation and enhancing LAB tolerance to acid through adaptive evolution. The future of LAB in beverage fermentation is bright, with advancements in technology, sustainability, and consumer-driven trends paving the way for innovative and health-enhancing products. Continued research and development will be crucial in overcoming current challenges and maximizing the benefits of LAB fermentation.”

Round 2
Reviewer 1 Report
Comments and Suggestions for Authors
Author revised the manuscript according to reviewer's comments. I have no further comments
Reviewer 2 Report
Comments and Suggestions for Authors
The authors have addressed my comments.